

# Kinematic response of ice-rise divides to changes in oceanic and atmospheric forcing

Clemens Schannwell[1], Reinhard Drews[1], Todd A. Ehlers[1], Olaf Eisen[2,3], Christoph Mayer[4], and
Fabien Gillet-Chaulet[5]

[1]Department of Geosciences, University of Tübingen, Tübingen, Germany
[2]Glaciology Section, Alfred Wegener Institute, Helmholtz Centre for Polar and Marine Research, Bremerhaven, Germany
[3]Department of Geosciences, University of Bremen, Bremen, Germany
[4]Bavarian Academy for Sciences and Humanities, Munich, Germany
[5]Univ. Grenoble Alpes, CNRS, IRD, Grenoble INP, IGE, Grenoble, France

**Correspondence:** Clemens Schannwell (Clemens.Schannwell@uni-tuebingen.de)

**Abstract.** The majority of Antarctic ice shelves are bounded by grounded ice rises. These ice rises exhibit local flow regimes that partially oppose the flow of the surrounding ice shelves. Formation of such ice rises is accompanied by a characteristic upward arching internal stratigraphy ("Raymond arches"), archiving potential past divide migration and the onset of divide flow. Information about past ice-sheet conditions can therefore be retrieved in areas where other archives are missing. However, the

quantitative interpretation of the stratigraphy requires modelling and radar observations. Hitherto, ice-rise modelling has been restricted to 2D and excluded the coupling between ice shelf and ice rise. This presents a major limitation for the interpretation of ice rises as ice-dynamic archive. Here we present an improved modelling framework to study ice-rise evolution using a satellite-velocity calibrated, isothermal, and isotropic 3D Full-Stokes model including grounding-line dynamics at the required mesh resolution (<500 m). We apply the model to the Ekström Ice Shelf catchment containing two ice rises. Our results

show that changes in the surface mass balance result in immediate and sustained divide migration (>2.0 m/yr) of up to 3.5 km. In contrast, instantaneous ice-shelf disintegration causes a short-lived and delayed (by 60-100 years) response of smaller magnitude (<0.75 m/yr). The model tracks migration of a triple junction and synchronous ice-divide migration in both ice rises with similar magnitude but differing rates. The model is suitable for glacial/interglacial simulations on the catchment scale, providing the next step forward to unravel the ice-dynamic history stored in ice rises all around Antarctica.

## 1  Introduction

Ice rises are parabolically-shaped surface expressions along the margin of the Antarctic ice sheet, and they form where the otherwise floating ice locally regrounds. They are characterised by their radial ice-flow centre - henceforth referred to as ice-rise divide - that is independent of the main ice sheet, resulting in divergence of the main ice flow around the obstacle. These obstacles act as a decelerating force that restricts ice flow which is commonly referred to as "ice-shelf buttressing". More than

700 ice rises (Matsuoka et al., 2015) are distributed along the Antarctic perimeter (Figure 1a), providing additional buttressing to the ice upstream. Ice rises archive their flow history in their characteristic internal stratigraphy (e.g. Conway et al., 1999;





Nereson and Waddington, 2002; Drews et al., 2015), making them potentially suitable sites for ice core drilling such as for the International Partnerships in Ice Core Sciences (IPICS) 2K and 40K Array (Brook et al., 2006).

Due to very low deviatoric stresses near the bed of the divide region and the power law rheology of ice, the effective viscosity (e.g. the stiffness of ice) is significantly higher towards the ice sheet bottom under the divide than in the surrounding areas. The

stiff ice impedes downward flow under the divide in comparison to the flank regions (Kingslake et al., 2016) such that ice of the same age is found at shallower depths in the ice column under the divide compared to the flank regions (Raymond, 1983). This results in the formation of upward arches in the isochronal ice stratigraphy commonly referred to as "Raymond arches".

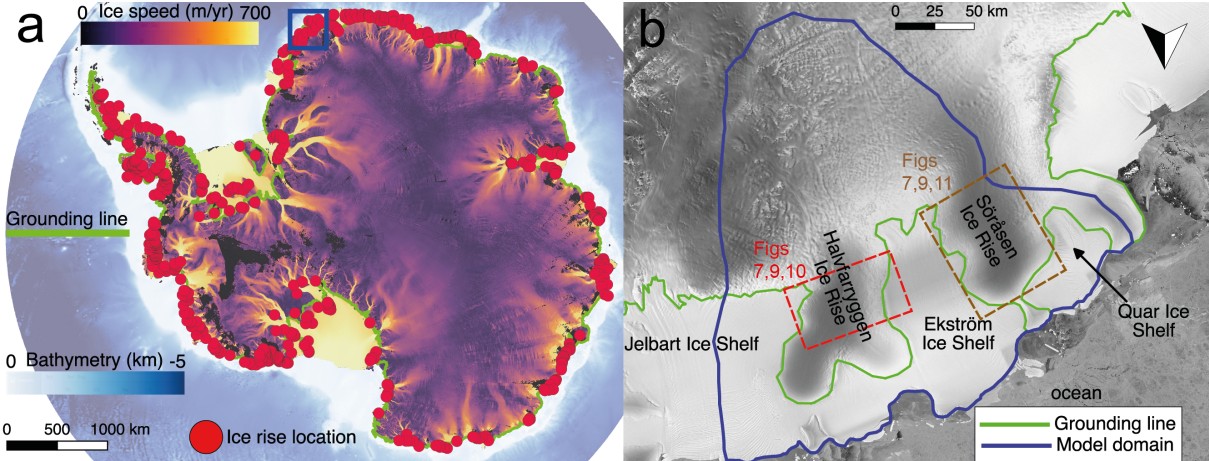

**Figure 1.** (a) Location map of ice rises along the margin of the Antarctic ice sheet (Matsuoka et al., 2015). The base map combines ice velocity of the ice sheet (Rignot et al., 2011) and the bathymetry of the adjacent ocean regions (Arndt et al., 2013). Blue rectangle shows zoom-in area of (b). (b) Radarsat image (Jezek and 5 RAMP-Product-Team:, 2002) of the study area with locations mentioned in the main text.

Previous studies have made much progress in interpreting the Raymond arches as ice-dynamic archive. The arch amplitude

and the tilt of multiple Raymond arches - commonly referred to as Raymond stack - have been used to infer migration of ice divides (e.g. Nereson et al., 1998), onset of local divide flow (e.g. Conway et al., 1999; Martín et al., 2006, 2014), and ice rise residence time (e.g. Drews et al., 2013, 2015; Goel et al., 2017). Common to all these studies is the comparison of the observed isochrones derived from radar data, with the predicted age fields from models with varying input scenarios. The full stress solution of the Stokes equations is necessary because both longitudinal and bridging stress gradients are important near

ice divides (e.g. Martín et al., 2009b; Gillet-Chaulet and Hindmarsh, 2011).

Matching simulated arch amplitudes with observed amplitudes remains challenging and several studies (e.g. Pettit et al., 2007; Martín et al., 2009a) could not reproduce observed arch amplitudes with the most widely-used ice flow law (Glen's flow law, Eq. 3). It appears higher Glen flow indices than $n>3$ are required (Martín et al., 2006; Pettit et al., 2007; Drews et al., 2015). Observations from the Greenland ice sheet seem to support these flow conditions (Gillet-Chaulet et al., 2011; Bons et al.,





2018). Moreover, an anisotropic rheology is needed to simulate double-peaked Raymond arches (Martín et al., 2009a) and ice anisotropy also impacts the arch amplitudes (Martín et al., 2014). Other challenges in modelling the internal stratigraphy include local minima in the surface mass balance (SMB) that are often not captured at the required detail in regional atmospheric models, but affect the modelled arch amplitude (Drews et al., 2015; Callens et al., 2016).

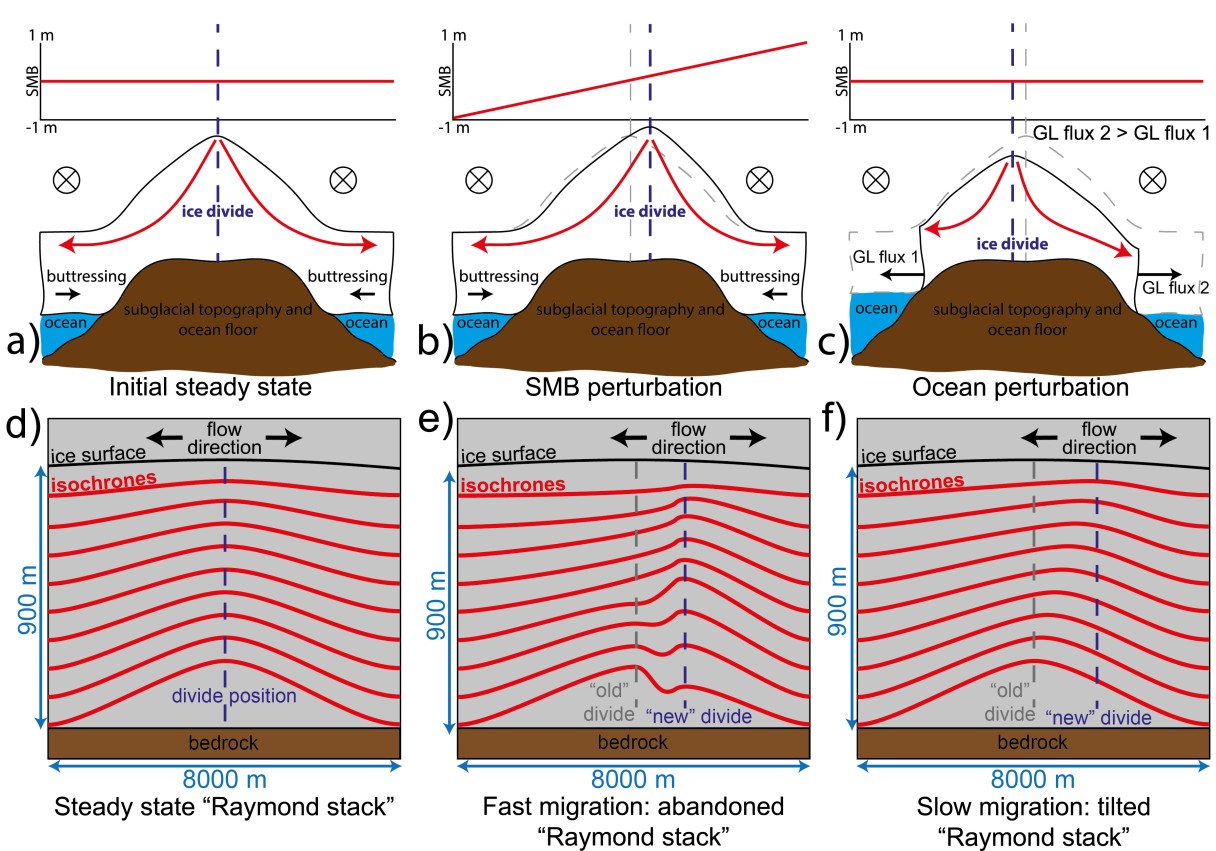

**Figure 2.** Upper panel (a-c) shows schematic steady state (a), divide migration induced by asymmetric surface mass balance forcing (b), and by ocean perturbation forcing (c). Buttressing in (c) is asymmetric. Solid red arrows indicate approximate ice flow path from ice-rise divide to ice shelf. Grey dashed lines in (b,c) display steady state geometry and divide position of (a). GL = grounding line. Lower panel (d-f) shows schematic of expected internal stratigraphy for steady state (d), fast migration (e), and slow migration (f).

Slow migration of ice divides results in tilted Raymond stacks (Nereson et al., 1998; Nereson and Waddington, 2002; Jacobson and Waddington, 2005; Martín et al., 2009b). Fast migration of ice divides results in abandoned Raymond stacks in the flanks and a new Raymond stack starts to develop at the new divide position (Jacobson and Waddington, 2005; Martín et al., 2009b, Figure 2d-f). For a new Raymond stack to form, a sudden displacement of 1-2 ice thickness is required (Martín
10 et al., 2009b), but thus far a clear threshold between both end-member scenarios remains elusive. Comparatively little is known about the stability of ice-divide triple junctions (Gillet-Chaulet and Hindmarsh, 2011). Triple junctions are points where three



ice-divide ridges meet. They often coincide with the summits of ice domes. It is unclear if some transient changes in such a triple-junction setting could explain observed relict arches in ice-divide flanks, which cannot be explained with ice-divide migration in a 2D setting (Drews et al., 2015). As ice divides and triple junctions are rarely perfectly elongated or axisymmetric, three dimensional effects cannot be neglected, necessitating the use of a 3D ice-sheet model. Even though divide migration has

been an active focus in ice-rise research, most studies have focussed on the ramifications of divide migration (Hindmarsh, 1996; Nereson et al., 1998; Jacobson and Waddington, 2005; Conway and Rasmussen, 2009), rather than what causes the divide to migrate. This is because the modelling has thus far largely been restricted to 2D and did not include interactions between ice rises and the surrounding ice shelves (Martín et al., 2009a, b; Gillet-Chaulet and Hindmarsh, 2011; Drews et al., 2015).

Here, we use the Full-Stokes (FS) ice-sheet model Elmer/Ice in 3D (Gagliardini et al., 2013) and extend the model domain in

contrast to previous studies to include grounding-line dynamics and ice-shelf flow to study potential causes for ice-rise divide migration. We apply the model to the Ekström Ice Shelf catchment bounded by two large ice rises. The model is calibrated by tuning basal friction and ice viscosity so that modelled surface velocities match today's flow field. Perturbations to the SMB and ice-shelf thickness are applied in forward simulations to investigate the coupled transient response of the two ice rises. The experiments are tailored to help improve our understanding of what processes cause ice-divide migration. Specifically,

we address the question of: is the amplitude of divide migration controlled by the SMB, ice-shelf buttressing, or is the divide position determined by the subglacial topography? Do ice rises in close proximity of each other show a similar response? Can we differentiate between the different trigger mechanisms? Furthermore, we investigate if the triple junction at one of the ice rises in the catchment - Halvfarryggen ice rise – also migrates in synchronicity with the main divide ridge

## 2   Study area: Ekström Ice Shelf catchment

Ekström Ice Shelf is located in the Atlantic sector of the East Antarctic ice sheet and is a medium-sized ice shelf. The embayment is characterised by two prominent ice rises: Söråsen ice rise in the west and Halvfarryggen ice rise, an ice promontory sandwiched between Ekström Ice Shelf and Jelbart Ice Shelf in the east (Figure 1b). While Söråsen ice rise is made up of a single ridge divide, Halvfarryggen consists of three main ridges that meet to form a triple junction close to the summit (Hofstede et al., 2013). Both ice rises belong to the larger ice rises in Antarctica with areas of about 5700 km2 and 5500 km2 for

Halvfarryggen and Söråsen, respectively (Matsuoka et al., 2015). Söråsen rce rise is additionally buttressed in the west by Quar Ice Shelf (Figure 1b). SMB across both divide ridges is strongly asymmetric with the western downwind side of the divide receiving much less accumulation (<0.6 m/yr ice equivalent) than the eastern side (up to 2.9 m/yr ice equivalent) (Drews et al., 2013; Lenaerts et al., 2014). The catchment is a suitable test site to study the dynamics of ice rises because the boundary input datasets such as subglacial topography, ice thickness, and ice surface elevation are well constrained (with many flightlines

across the region due to the vicinity of the Neumayer III airfield). In addition, the confined geometry of both ice rises suggests that they receive significant lateral buttressing by Ekström and Jelbart ice shelves. This means that the buttressing Ektröm and Jelbart ice shelves receive also restricts ice flow across the grounding line from both ice rises, making the divide position potentially sensitive to ocean forcing.





## 3 Methods

### 3.1 Model description

#### 3.1.1 Governing equations

Ice flow is dominated by viscous forces (e.g. low Reynolds number) permitting the dropping of the inertia and acceleration
terms in the momentum equations. Using these simplifications, the most complete description of ice flow is the FS flow model.
The Stokes equations for linear momentum are

$$\mathrm{div}\boldsymbol{\sigma} = \rho_{\mathrm{i}}\boldsymbol{g}, \tag{1}$$

where $\boldsymbol{\sigma} = \boldsymbol{\tau} - \mathrm{p}\boldsymbol{I}$ is the Cauchy stress tensor, $\boldsymbol{\tau}$ is the deviatoric stress tensor, $\mathrm{p} = -\mathrm{tr}(\boldsymbol{\sigma})/3$ is the isotropic pressure, $\boldsymbol{I}$ the
identity matrix, $\rho_{\mathrm{i}}$ the ice density, and $\boldsymbol{g}$ is the gravitational vector. Ice flow is assumed to be incompressible which simplifies
mass conservation to

$$\mathrm{div}\boldsymbol{u} = 0, \tag{2}$$

with $\boldsymbol{u}$ being the ice velocity vector. Here we model ice as an isotropic material. Its rheology is given by Glen's flow law which
relates deviatoric stress $\boldsymbol{\tau}$ with the strain rate $\dot{\boldsymbol{\epsilon}}$:

$$\boldsymbol{\tau} = 2\eta\dot{\boldsymbol{\epsilon}}, \tag{3}$$

where the effective viscosity $\eta$ can be expressed as

$$\eta = \frac{1}{2}\mathrm{EB}\dot{\boldsymbol{\epsilon}}_e^{\frac{(1-\mathrm{n})}{\mathrm{n}}}. \tag{4}$$

In this equation E is the enhancement factor, B is a viscosity parameter computed through an Arrhenius law, n is Glen's flow
law parameter (n=3), and the effective strain rate is defined as $\dot{\boldsymbol{\epsilon}}_e^2 = \mathrm{tr}(\dot{\boldsymbol{\epsilon}}^2)/2$. Although there is evidence that the Glen flow
parameter is >3 (Bons et al., 2018), particularly near ice rises (Gillet-Chaulet et al., 2011), we stick to the current modelling
standard of n=3, because we focus on divide migration rather than Raymond arch amplitudes here. The same holds for the
enhancement factor, which is often used to account for anisotropic effects, but is set to 1 (isotropic conditions) in all simulations.
In future applications of this model, these assumptions should be revisited.



**Table 1.** Standard physical and numerical parameters used for the simulations

| Parameter | Symbol | Value | Unit |
|---|---|---|---|
| Gravity | $\boldsymbol{g}$ | 9.8 | m s$^{-2}$ |
| Ice density | $\rho_\mathrm{i}$ | 917 | kg m$^{-3}$ |
| Sea water density | $\rho_\mathrm{w}$ | 1028 | kg m$^{-3}$ |
| Glen's exponent | n | 3 | |
| Enhancement factor | E | 1 | |
| Basal friction exponent | m | 1 | |
| Basal melting tuning parameter I | G | 0.01 | |
| Basal melting tuning parameter II | A | 0.1 | |
| Basal melting tuning parameter III | $\alpha$ | 0.4 | |
| Ice temperature | T | -10 | °C |
| Simulation length | | 1000 | yr |
| Time step size | | 0.5 | yr |
| Maximum mesh refinement | | 500 | m |

### 3.1.2 Boundary conditions

A kinematic boundary condition determines the evolution of upper and lower surfaces $z_\mathrm{j}$

$$\frac{\partial z_\mathrm{j}}{\partial t} + u_\mathrm{x}\frac{\partial z_\mathrm{j}}{\partial x} + u_\mathrm{y}\frac{\partial z_\mathrm{j}}{\partial y} = u_\mathrm{z} + \dot{a}_\mathrm{j}, \tag{5}$$

where $\dot{a}_\mathrm{j}$ is the accumulation/ablation term and $\mathrm{j} = (\mathrm{b},\mathrm{s})$, with s being the upper surface and b being the lower surface (base)
of the ice sheet. The ice-shelf basal mass balance ($\dot{a}_\mathrm{b}$) parameterisation follows Favier et al. (2016)

$$\dot{a}_\mathrm{b} = H^\alpha(\rho_\mathrm{f}G + (1 - \rho_\mathrm{f})A), \tag{6}$$

where H is the ice thickness, G, A, and $\alpha$ are tuning parameters (Table 1). The parameter $\rho_\mathrm{f}(x,y)$ decreases exponentially with distance from the grounding line, varying from $\rho_\mathrm{f}(x,y) = 1$ at the grounding line and $\rho_\mathrm{f}(x,y) = 0$ some distance ($\sim$40 km) away from it. This ensures that highest melt rates are always specified at the grounding line and decrease exponentially with
10 distance away from it. The tuning parameters are chosen such that basal melt rates roughly agree with melt rates derived from satellite observations and mass conservation (Neckel et al., 2012). Melt rates under the shelf are generally low, ranging from -0.2 m/yr to 1.1 m/yr near the grounding line (Neckel et al., 2012). No basal melting is applied to the grounded ice sheet. Where the ice is in contact with the bedrock a linear Weertman-type basal sliding law is employed

$$\boldsymbol{\tau_b} = C|\boldsymbol{u_b}|^{\mathrm{m}-1}\boldsymbol{u_b}, \tag{7}$$

where $\boldsymbol{\tau_b}$ is the basal traction, m is the basal friction component – set to 1 in all simulations, and C is the basal friction coefficient inferred by solving an inverse problem (see section 3.3 Model initialisation). Underneath the floating part (ice



shelves) of the domain basal traction is zero ($\tau_b = 0$), but sea pressure is prescribed.

For the ice shelf front boundary, the true vertical distribution of the hydrostatic water pressure is applied and the calving front is held fixed throughout the simulations. We assume the ice is isothermal with a constant temperature of -10°C. We specify depth-independent horizontal ice velocities taken from the MEaSUREs datatset (Rignot et al., 2011) at all lateral boundaries

other than the ice-shelf front. To solve the presented system of equations with the appropriate boundary conditions, we use the open-source finite element code Elmer/Ice (Gagliardini et al., 2013).

### 3.2    Initial geometry and input data

To initialise the model geometry, subglacial topography was taken from the BEDMAP2 dataset (Fretwell et al., 2013). Our surface elevation is a merged product of Cryosat and, where available, higher resolution TanDEM-X digital elevation models

(V. Helm, pers. communication 2018). SMB ($\dot{a}_s$, Eq. 5) in our simulations was taken from regional climate model simulations with RACMO2.3 (Lenaerts et al., 2014). This model reproduces the commonly observed asymmetric SMB across ice rises (Callens et al., 2016). Velocity observations to match modelled ice velocities in the inversion procedure were taken from the MEaSUREs datatset (Rignot et al., 2011).

### 3.3    Model initialisation

We perform the model initialisation in two steps using two different mechanical models: the shallow shelf approximation (SSA;Morland (1987)) and the FS model. The SSA ice-sheet model is initialised by solving an optimisation problem to simultaneously infer the basal traction coefficient C and the viscosity parameter B. This type of snapshot initialisation is well known and widely employed in ice-sheet modelling and aims to match modelled velocities with observed velocities (e.g. MacAyeal, 1993; Gillet-Chaulet et al., 2012; Cornford et al., 2015). In a more formal way, we seek a minimum of the objective function

$$J = J_m + J_p,\tag{8}$$

where $J_m$ is the misfit between observed and modelled velocities and $J_m$ is a Tikhonov penalty function described by

$$J_p = \lambda_C J_C^{reg} + \lambda_B J_B^{reg},\tag{9}$$

where $\lambda_C$ and $\lambda_C$ are the Tikhonov parameters $J_C^{reg}$ and $J_C^{reg}$ represent the smoothness constraints of basal drag and viscosity (e.g. Gillet-Chaulet et al., 2012; Cornford et al., 2015). The smoothness constraints penalise the square of the first derivatives.

An L-curve analysis was performed to calibrate the Tikhonov parameters and avoid overfitting or overregularisation (Fürst et al., 2015). The selected values are $\lambda_C = 10^7$ and $\lambda_B = 10^6$ (Figure 3).

However, when the inferred basal traction coefficient and viscosity fields from the SSA inversion were used in forward simulations, unrealistically high surface lowering rates (200 m/century) in the vicinity of the divide were observed. To circumvent this problem, the output fields from the SSA inversion were used as input for a second inversion using the FS model, but

only adjusting the basal friction coefficient, while keeping the viscosity of the first inversion fixed. The Tikhonov parameter from the SSA inversion was used for the FS inversion as well. For both inversions a horizontal grid resolution of ~1 km was





used. Following initialisation, as is commonly done in ice-sheet modelling, we performed a short 10-year relaxation simulation to smooth out data inconsistencies. This simulation length lies in the range of what other studies have done previously (e.g. Cornford et al., 2015; Schannwell et al., 2018).

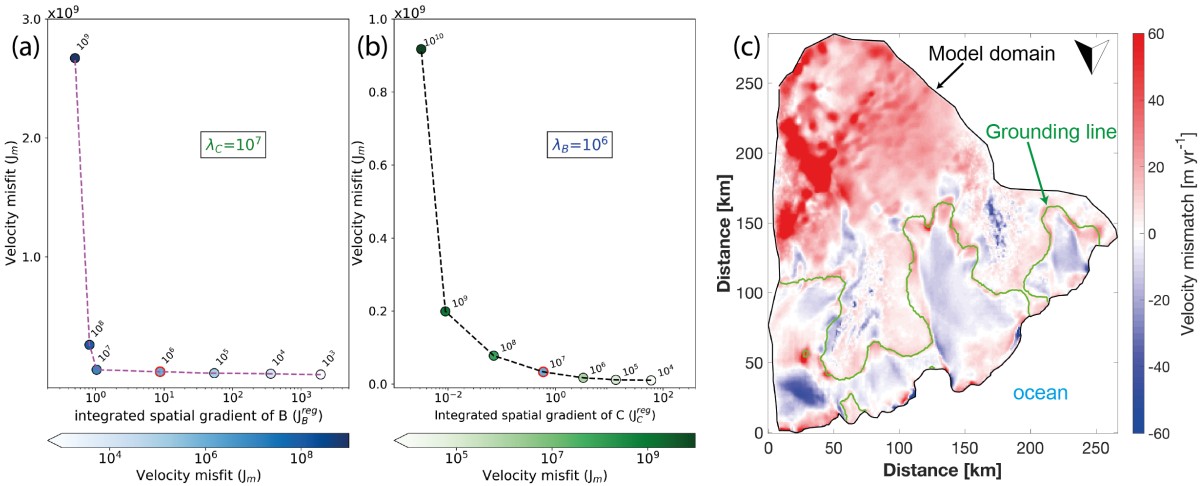

**Figure 3.** L-curve analysis to select Tikhonov parameters $\lambda_B$ and $\lambda_C$: (a) 2-D cross section for variable $\lambda_B$ and $\lambda_C$ fixed at $10^7 \text{Pa}^{-2}\text{m}^6\text{a}^{-4}$. (b) Reverse case with constant $\lambda_B$ at $10^6 \text{m}^4\text{a}^{-2}$ and varying $\lambda_C$. The units of $J_m$ and $J_C^{\text{reg}}$ are $\text{m}^4\text{a}^{-2}$ and $\text{Pa}^2\text{m}^{-2}\text{a}^2$, respectively. $J_B^{\text{reg}}$ is unitless. (c) shows mismatch between modelled and observed velocities (Rignot et al., 2011) after FS inversion for the modelling domain (Figure 1b, blue outline).

## 3.4 Experimental design

The forward simulations focus on two types of perturbation simulations (1) perturbations to the SMB, and (2) ocean perturbations resulting in changing ice-shelf thickness and hence ice-shelf buttressing. For the SMB, we use modelled values from RACMO2.3 (Lenaerts et al., 2014). The spatial pattern of the SMB is such that low accumulation rates are applied on the western (downwind) side (<0.5 m/yr) of the ice rises and high accumulation rates (>1 m/yr) are applied on the eastern (upwind) sides. This asymmetric SMB pattern is consistent with observations (Drews et al., 2013), but does not capture the correct

magnitudes. Therefore, we adjust the SMB using the observed model drift following the relaxation simulation. The SMB is held constant in time, and the adjustment is equal to the computed spatial thickening/thinning rate and keeps all ice divides at the initial position. This is our reference simulation, and we treat the unadjusted SMB as a simulation with a perturbed SMB. In the second type of experiments, we perturb the reference run by thinning ice shelves through an increase in ocean induced melting. The extreme scenario, where all ice shelves are removed, is simulated by cutting the numerical mesh ∼1km

downstream of the present grounding-line position. This ensures that the same frontal boundary conditions still apply to this geometry. To test more ice-shelf buttressing reduction scenarios, we performed additional model simulations with intermediate shelf thinning scenarios, where shelf thickness was reduced by 10% and 50% at the start of the perturbation simulation. For both simulations the shelf geometry is kept constant for the remainder of the simulation by applying a synthetic steady-state





ocean forcing. To permit a more direct comparison between ocean forcing and SMB forcing, an additional SMB perturbation simulation is performed, where the SMB is unadjusted and the initial grounding-line flux perturbation from the shelf removal simulation on either side of Halvfarryggen is added to the SMB term. This is done such that the spatial pattern of the SMB remains unchanged, but the magnitude is different by about a factor two in comparison to the unadjusted SMB. All pertur-

5 bation simulations are run forward for 1000 years which is about the characteristic time T (T=ice thickness/accumulation) for Halvfarryggen and Söråsen ice rises (Drews et al., 2013). After one characteristic time, the formation of synclines in the internal stratigraphy can usually be observed (Martín et al., 2014). Most of the simulations are performed with two different mesh resolutions (Figure 4). The first isotropic mesh uses a horizontal resolution of ∼2 km throughout the domain (henceforth regular mesh), whereas for the second mesh (henceforth refined mesh), we initially use the same footprint as for the regular

10 mesh, but use the meshing software MMG (http://www.mmgtools.org/), to refine the mesh along the grounding line and all ice divide ridges to a resolution of ∼500 m. Mesh resolution then decreases with distance away from the regions of interest to the lowest resolution of ∼10 km (Figure 4). A second refined mesh, where we refine down to ∼350 m at the divides and grounding line, was used for one simulation. All meshes are held fixed over time and no dynamic remeshing is performed. A summary of all perturbation experiments is provided in Table 2.

**Table 2.** List of all perturbation experiments including forcings and mesh resolution

| Run # | Ocean forcing | SMB forcing | Mesh (max. resolution) |
|:---:|:---:|:---|:---:|
| 1 | Function of distance to GL (Eq. 6) | Adjusted | Regular mesh (2 km) |
| 2 | Function of distance to GL (Eq. 6) | Adjusted | Refined mesh (0.5 km) |
| 3 | Function of distance to GL (Eq. 6) | Unadjusted | Regular mesh (2 km) |
| 4 | Function of distance to GL (Eq. 6) | Unadjusted | Refined mesh (0.5 km) |
| 5 | Function of distance to GL (Eq. 6) | Unadjusted | Refined mesh (0.35 km) |
| 6 | Function of distance to GL (Eq. 6) | Unadjusted + flux perturbation Run 6 | Refined mesh (0.5 km) |
| 7 | No shelf + function of distance to GL (Eq. 6) | Adjusted | Regular mesh (2 km) |
| 8 | No shelf + function of distance to GL (Eq. 6) | Adjusted | Refined mesh (0.5 km) |
| 9 | 90% shelf thickness + adjusted ocean forcing | Adjusted | Refined mesh (0.5 km) |
| 10 | 50% shelf thickness + adjusted ocean forcing | Adjusted | Refined mesh (0.5 km) |





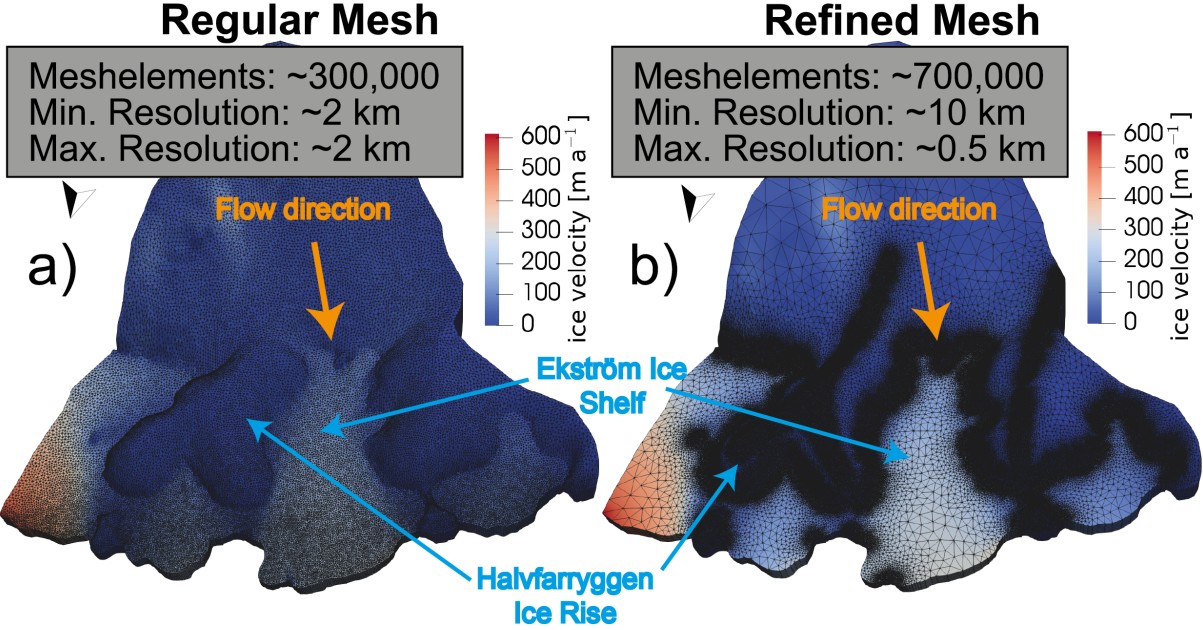

**Figure 4.** 3D plot of the model domain showing the two main mesh resolutions employed in this study. (a) shows uniform mesh of 2 km resolution and (b) shows refined mesh in areas of interest such as the grounding line and the ice rise divides. In these areas mesh resolution is ∼500 m and away from these regions mesh resolution increases to 10 km.

## 4  Results

### 4.1  Effect of mechanical model in initialisation on transient simulations

We observed an order of magnitude difference in the inverted basal drag coefficients, depending on whether we use the SSA or the FS approximation as forward model (Figure 5). This means that sliding is more restricted in the FS case in comparison to SSA case. Most notably, in comparison to the SSA inversion, the slippery regions in the FS inversion do not extend as far upstream. For example, between the ice divide of Halvfarryggen and the grounding line on either side, there are areas of stickier bedrock conditions that are missing in the drag field of the SSA inversion (Figure 5). Spuriously high ice surface lowering rates (200 m/century) in divide proximity were simulated when basal drag and viscosity fields were used from the inversion using the SSA. Thinning rates were much smaller when using the FS forward model.





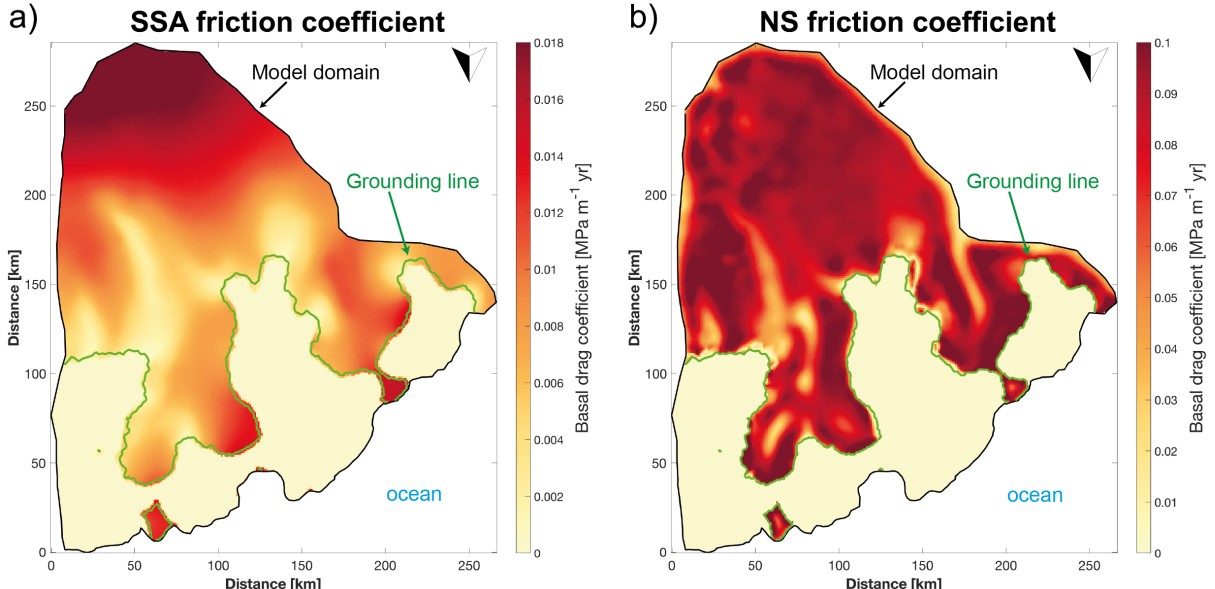

**Figure 5.** Inferred basal traction fields C using the SSA model (a) and the FS model (b). Note the difference in the colourbar scale.

### 4.2 Reference simulation (Runs 1-2)

The reference simulation serves the purpose of ensuring that the applied synthetic SMB does indeed result in a steady-state geometry in which divide positions do not migrate and surface topography changes are minimal. It then also functions as a baseline against which the perturbation simulations can be compared. The thinning/thickening rates in this type of simulation

for both meshes are highest at the start of the simulation, but never exceed 0.05 m/yr near the divide region. These low thinning/thickening rates result in a steady state geometry of the model domain that is also characterised by stable positions of the ice rise divides. For both ice rises total divide migration amplitudes are <60 m throughout the forward simulation.

### 4.3 Surface mass balance induced divide migration (Runs 3-6)

#### 4.3.1 Halvfarryggen ice rise

In simulations 2-4 (Table 2), the SMB perturbation results in immediate divide migration for both meshes (Figure 6a). For Halvfarryggen, we focus our analysis on the main (eastern) divide ridge (Figure 1b). Owing to the more positive SMB on the eastern side of both divides, the divide migrates towards this region (Figure7a,b). Almost all of the divide migration occurs over the first 200 years of the simulation before a new steady-state position is reached (Figure 6a). During the first 200 years, the entire divide migrates at an average rate of 16-20 m/yr to the east for Halvfvarryggen ice rise. This range shows there is a clear

mesh dependence on the magnitude of divide migration over this timeframe (3.2-3.8 km, Figure 6a), whereas this difference is less pronounced in the steady-state divide positions at the end of the simulation (2.5-2.8 km). For the two refined meshes (Runs 4,5; Table 2), mesh dependence is still present, even if reduced, and first order convergence between the simulations



is absent. This indicates that very fine mesh resolution is required to capture divide migration, but in the light of the high computational costs to run all simulations at such a resolution, we restrict ourselves to a maximum resolution of 500 m. While the averaged absolute magnitude along the swath profile is offset by about 300 m between the refined grid simulations (Runs 3,4; Table 2), the temporal pattern of divide migration is identical (Figure 6a). This is in contrast to the regular mesh, where

divide migration reaches its maximum (most eastern position) after ∼200 years and remains almost stable for the remaining simulation period. In comparison, the refined mesh simulations reach their maximum divide migration at a similar time (∼200 years), but start to slowly migrate back towards its initial position until a steady state is reached after 700 years (Figure 6a). The SMB grounding-line flux perturbation simulation (Run 6, Table 2) has almost an identical steady state as the unadjusted SMB simulation (Figure 6a), despite the SMB flux disparity between east and west of the divide being twice as high. While this

does not affect the steady-state divide position, it does result in faster migration with a larger maximum amplitude of divide migration (∼3.9 km vs. ∼3.4 km).

### 4.3.2    Söråsen ice rise

Söråsen ice rise shows a very similar response – both in absolute magnitude of divide migration as well as temporal evolution of divide migration - to the applied SMB perturbation (Figure 6b) The most pronounced differences of ∼4 m/yr are in the

divide migration rates (11-15 m/yr) in the first 200 years of the simulation, when most of the divide migration takes place. The mesh dependence on divide-migration amplitudes is also present for Söråsen, even though it is slightly reduced in comparison to Halvfarryggen (2.9-3.3 km, Figure 6b). Unlike Halvfarryggen, simulations with the refined mesh do not show any backward migration of the Söråsen divide (Figure 7c,d), but reach a new steady-state position after ∼300-400 years with little migration beyond this point in all simulations (Figure 6b).

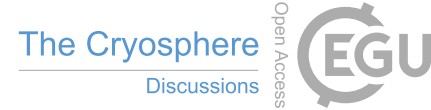

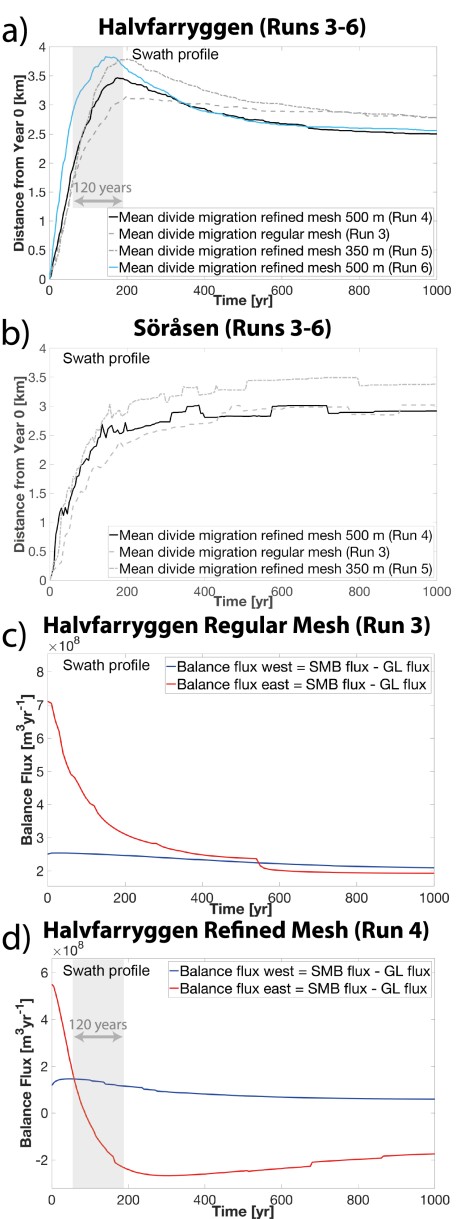

**Figure 6.** Ice-rise divide migration for Halvfarryggen ice rise (a) and Söräsen ice rise (b) induced by surface mass balance perturbation for different mesh resolutions. Positive numbers indicate migration to the east. (c,d) show balance fluxes for the eastern and western side of the divide to the grounding line for Halvfarryggen ice rise. Grey shaded areas in (a) and (d) highlight time lag between balance flux east being smaller than corresponding flux west and the start of backward migration. "Swath profile" indicates that computed divide migration is averaged over cross-sections shown in Figure 7. For the grounding line (GL) flux, this means that the flux is summed along the current grounding-line position over the swath length. The surface mass balance (SMB) flux is area averaged from the current divide position to the respective grounding-line position in the east and west of the divide over the swath length. Note different y-axis scales in (c) and (d).





**Figure 7.** Ice-rise divide positions at selected times for Halvfarryggen ice rise (a,b) and Söräsen ice rise (c,d) induced by surface mass balance perturbation for the regular mesh (a,c) and the refined mesh (b,d).





### 4.4 Ocean perturbation induced divide migration (Runs 7-10)

Ocean perturbation causes the adjacent ice shelves to thin and reduce their ability to buttress the ice upstream. If buttressing of ice shelves on either side of the ice rise is asymmetric, this should lead to an asymmetric increase in grounding-line flux, which in turn should result in an asymmetric thinning perturbation. This asymmetric thinning perturbation should then "push" the divide in the direction of the smaller thinning perturbation (Figure 2c). In our model simulations, we test this hypothesis using the real-world example of the Ekström Ice Shelf embayment.

#### 4.4.1 Halvfarryggen ice rise

All ocean perturbation experiments result in an instantaneous divide migration. For the extreme case of ice-shelf removal (Runs 7-8, Table 2) the maximum ice-divide migration is -0.4 km and -0.7 km and occurs after 700 years and 1000 years for the regular and refined mesh, respectively (Figure 8a). Negative numbers indicate westward migration (Figure 9a,b). For the intermediate thinning scenarios (Runs 9-10, Table 2), there is no divide migration for the 10% thinning scenario and -0.2 km divide migration for the 50% thinning scenario. So even though half of the shelf's thickness is removed, the magnitude of the divide migration is reduced by 77%, indicating that the response to ice-shelf thinning is non-linear and requires a strong perturbation for a significant response. Most of the divide migration takes place in the centuries following the perturbation. A stable divide position is reached after ~690 years for the regular mesh simulation and after ~480 years for the refined mesh simulation in the shelf removal scenario (Figure 8a). In the 50% shelf thinning scenario, a stable divide position is reached after ~600 years. As was the case for the SMB, the shelf removal simulations exhibit mesh dependence with the refined mesh increasing divide migration by 69% (Figure 8a).

To shed light on why divide migration amplitudes differ for the different shelf thinning scenario, grounding-line fluxes for the eastern and western side were computed for Halvfarryggen (Figure 8c,d and Figure 10c). In all simulations, an initial sharp increase in grounding-line flux is computed, which then quickly decays and is even lower for the eastern side of the divide than in the reference simulation (Figure 8c,d). The initial flux perturbation is largest for the shelf removal scenario and is non-linear, where halving of the ice-shelf thickness results in a flux reduction of 60%.



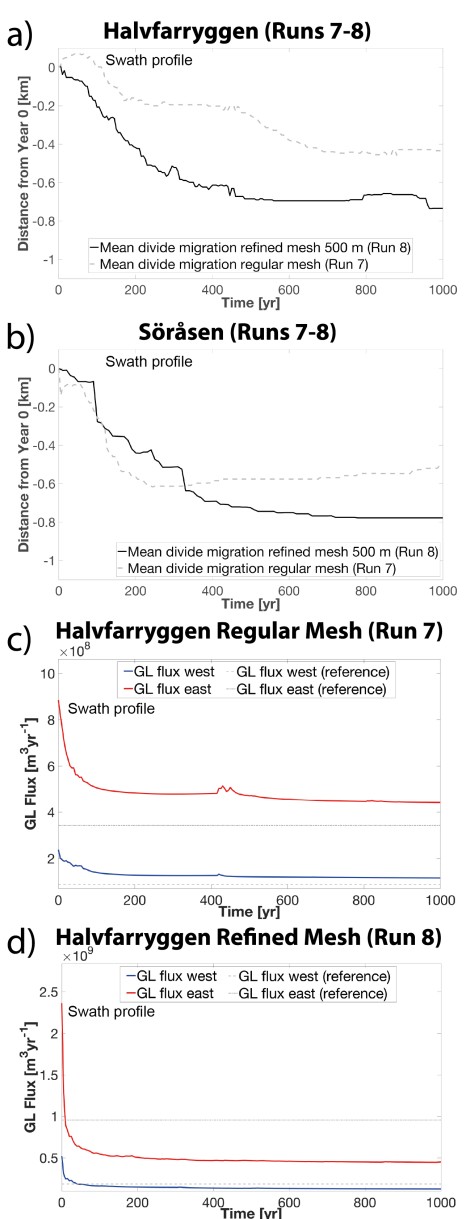

**Figure 8.** Ice-rise divide migration for Halvfarryggen ice rise (a) and Söräsen ice rise (b) induced by ocean perturbation (shelf removal) for different mesh resolutions. Negative numbers indicate migration to the west. (c,d) show grounding-line (GL) fluxes for the eastern and western side of the divide to the grounding line for Halvfarryggen ice rise. "Swath profile" indicates that computed divide migration is averaged over cross-sections shown in Figure 9. For the GL flux, this means that the flux is summed along the current grounding-line position over the swath length. Note different y-axis scales in (c) and (d).





**Figure 9.** Ice-rise divide positions at selected times for Halvfarryggen ice rise (a,b) and Söräsen ice rise (c,d) induced by ocean perturbation (shelf removal) for the regular mesh (a,c) and the refined mesh (b,d).





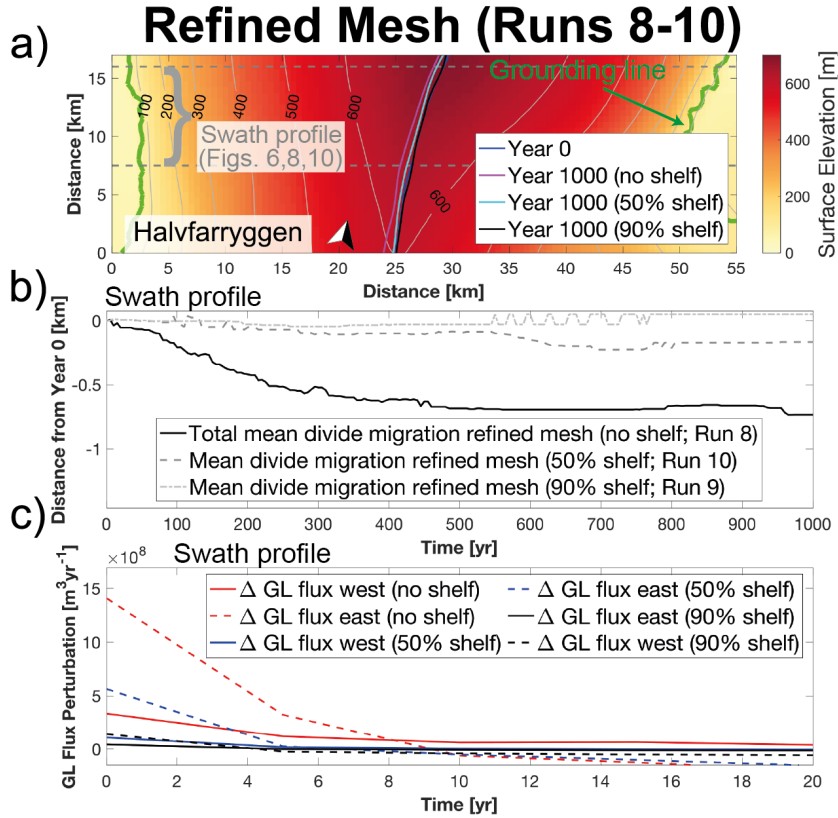

**Figure 10.** Ice-rise divide migration for Halvfarryggen ice rise induced by different ocean perturbations (shelf removal, 50% ice-shelf thickness, and 90% ice-shelf thickness) for the refined mesh (a-c). (a) shows divide position at the end of the simulation period. Grey dashed lines approximate area used for flux calculations in (c). (b) shows mean divide migration for the different perturbations. Negative numbers indicate migration to the west. (c) displays ice flux perturbation across the grounding line for the eastern and western side of the divide for the first two decades. Perturbation fluxes smaller than 0 indicate that ice flux across the grounding line is reduced in comparison to the reference simulation. (b,c) "Swath profile" indicates that computed divide migration is averaged over cross-sections as shown in (a). For the ΔGL flux, this means that the flux is summed along the current grounding-line position over the swath length.

### 4.4.2  Söråsen ice rise

For the extreme case of ice-shelf removal (Runs 7-8,Table 2) the maximum ice-divide migration is -0.5 km and -0.7 km for the regular and refined mesh, respectively (Figure 8b). Almost all of the divide migration happens in the first half of the simulation period before a new steady-state position is reached after 300 years for the regular mesh and after ∼400 years for the refined mesh (Figure 8b). This behaviour closely follows Halvfarryggen both in terms of migration amplitudes and direction (Figures 8b, 9c,d). This is not the case for the intermediate thinning scenarios (Runs 9-10, Table 2), where no divide migration (<60 m) occurs in any of them.



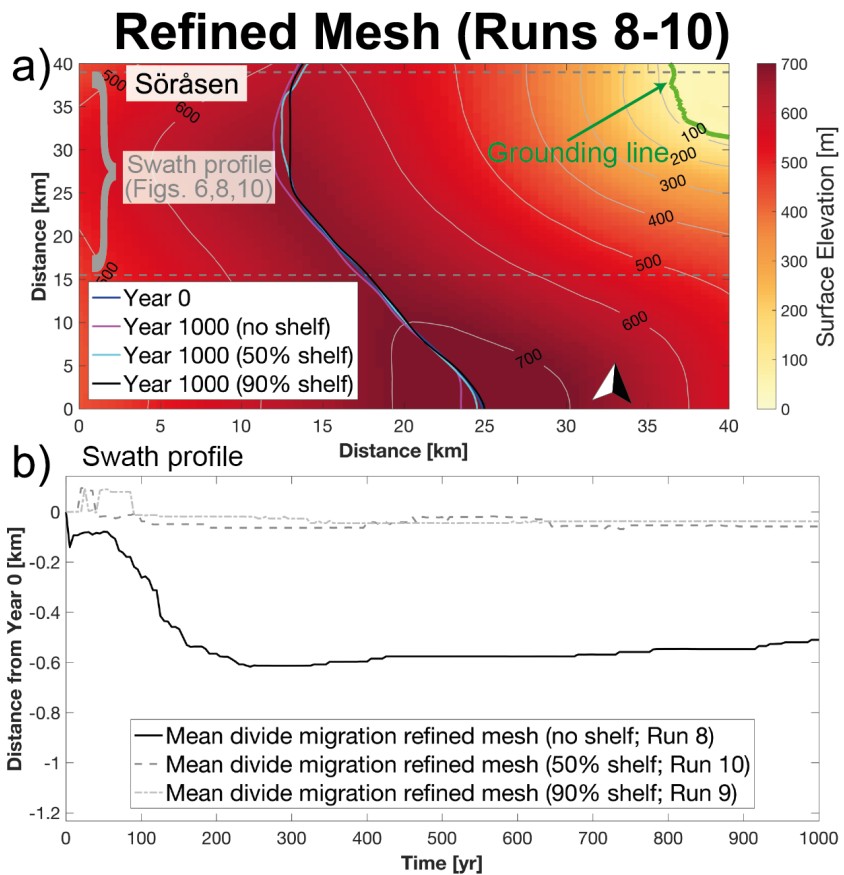

**Figure 11.** Ice-rise divide migration for Söråsen ice rise induced by different ocean perturbations (shelf removal, 50% ice-shelf thickness, and 90% ice-shelf thickness) for the refined mesh (a,b). (a) shows divide positions at the end of the simulation period. (b) shows mean divide migration for the different perturbations. Negative numbers indicate migration to the west.

### 4.5 Triple junction migration Halvfarryggen ice rise

In both experiments (Runs 4,8), the triple junction immediately migrates in response to the perturbations. The triple junction evolution closely follows the temporal evolution of the main divide. The maximum migration amplitude is 3.3 km and -1.2 km for the SMB and shelf removal simulations, respectively. Most of the triple junction migration takes place over the first ∼400

5    years, before a new steady-state position is reached (Figure 12). The temporal evolution of the triple junction migration also shows a tendency to migrate back to its initial position in the latter part of the SMB simulation (Figure 12c). Because the SMB is most positive to the east of the main divide arm, the eastward migration of the divide leads to an increased angle between these two ridges of 3.5°, translating into a widening of 5%. In addition to the widening, the minor ridge appears more less distinct at the end of the simulation period (Figure 12b). This feature is absent in the shelf removal simulation. In contrast to

10    the SMB simulation, the westward migration of the divide leads to a narrowing of the angle between the two ridges, albeit only



by 1.1°, corresponding to a narrowing of 1.4%. The main component of the migration is east-west, but in both simulations the triple junction also migrates south.

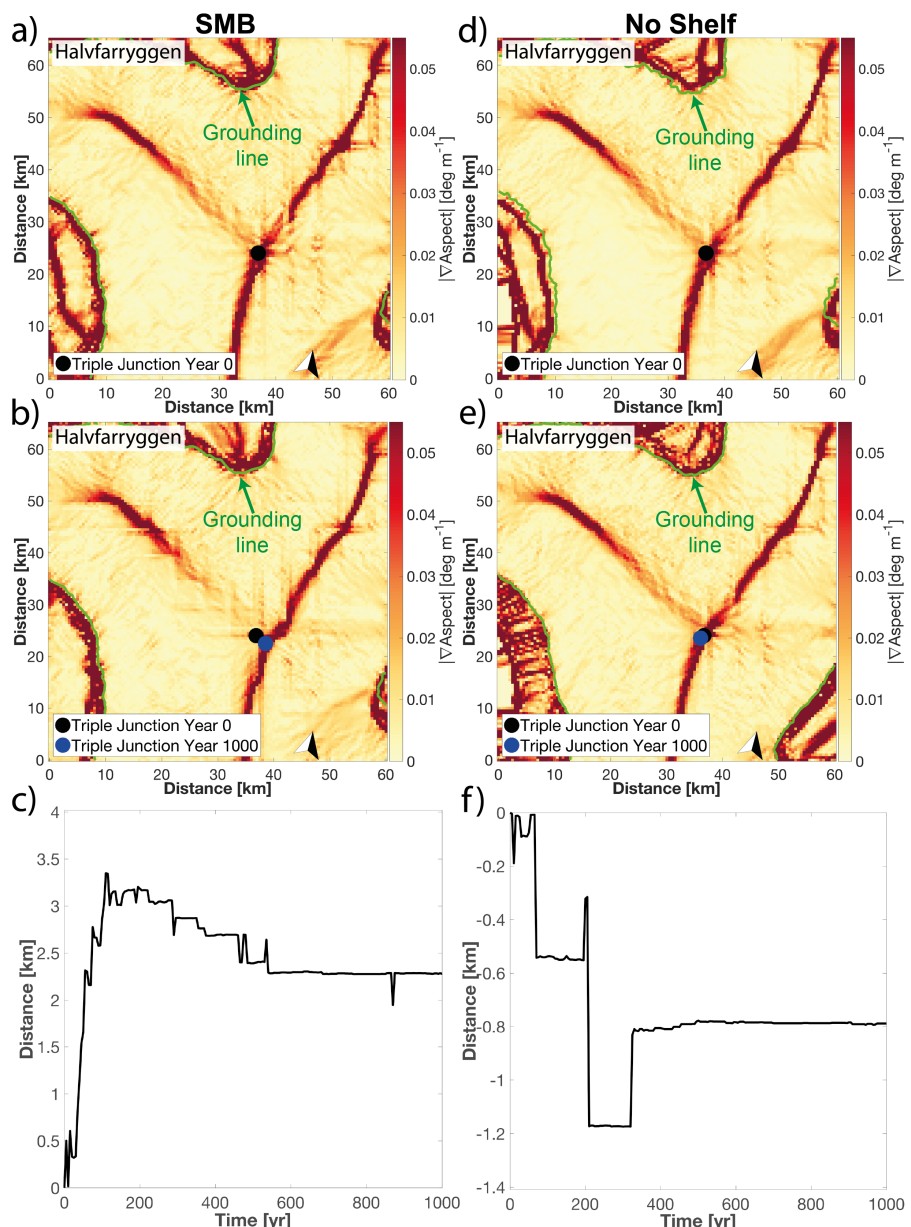

**Figure 12.** Triple junction migration for Halvfarryggen ice rise induced by surface mass balance perturbation for the refined mesh (a-c) and ocean perturbation (shelf removal) for the refined mesh (d-f). Upper and middle panels (a,b,d,e) show triple junction position at the start and end of the simulation period, respectively. Lower panel (c,f) shows mean divide migration for the respective perturbation simulations. Negative numbers indicate migration to the west. Note different y-axis scales in (c) and (f).





## 5 Discussion

### 5.1 Effect of mechanical model in initialisation on transient simulations

Transient simulations using basal drag coefficient and ice viscosity fields from the SSA inversion result in unrealistically large ice mass loss rates in the decades following model initialisation. Knowing that the Ekström catchment is likely close to a steady-state (Drews et al., 2013), we attribute these differences in the transient simulations to the difference in force balance approximation used in the inversion and transient simulation. This shows that for the presented ice-rise modelling a FS inversion is necessary for plausible transient simulations.

### 5.2 Surface mass balance induced divide migration

The computed mean divide-migration rates of 2.5-3.5 m/yr for both ice rises are higher than what has previously been inferred from the geometric analysis of Raymond stacks (0.5 m/yr Siple Dome; Nereson et al. (1998)). This means a realistic SMB perturbation, which could have occurred at the Last Glacial Macximum (LGM), leads to fast divide migration. Even more so because these migration rates are even higher (11-20 m/yr), if only the first 200 years are considered. Such a perturbation would likely lead to an abandoning of the Raymond stack and the formation of a new Raymond stack at the new steady-state divide position (Figure 2e), as the divide migrates >3 ice thicknesses away from its initial position. Because the Raymond stack at Halvfarryggen and Söråsen are well developed and it takes ∼10T (T = characteristic time, T=900 years (Drews et al., 2013)) to form such a stack, we conclude that both ice rises have not experienced a perturbation of such magnitude over the last ∼9000 years, indicating stable ice-flow conditions in this embayment for at least the Holocene time period.

For both ice rises, divide migration shows a clear dependence on mesh resolution. Not only are migration amplitudes different, but for Halvfarryggen a finer mesh resolution also results in differences in temporal divide migration pattern, with refined mesh simulations showing a subtle backward migration trend in the latter half of the simulation period (Figure 6a). We attribute this backward migration pattern to be a direct result of an imbalance in balance fluxes for the eastern and western side of the divide. In the regular mesh simulation (Run 3, Table 2), the more positive SMB on the eastern side of the divide results in an initial increase in balance flux, which slowly decays over time because the grounding-line flux compensates for the increased ice thickness by discharging more ice into the shelf. After ∼400 years the balance flux on the eastern side is lower than the corresponding flux on the western side, but the difference is small and does not result in any changes in divide position (Figure 6c). For the refined mesh simulations (Run 4, Table 2), the same initial increase in balance flux is computed. But the decay occurs much quicker than in the regular mesh simulation, showing that the grounding-line flux compensates much quicker for the increased ice thickness (Figure 6d). This leads to the eastern balance flux being lower than the corresponding western balance flux after ∼80 years. However, the balance flux continues to decrease because of the continuing increase in grounding-line flux up to ∼250 years, before it recovers slightly. The negative balance flux in the east leads to the computed subtle back migration trend from ∼185 years to ∼700 years. However, the timing of when the balance flux in the east starts to be lower than the balance flux in the west is 120 years prior to this (Figure 6a,d). This means that there is a time lag of 120 years before the divide reacts to changes in the balance fluxes. The time lag may be a little smaller since the balance flux in the




east is also lower than the balance flux in the west for the regular mesh simulation, but does not result in backward migration of the divide. This indicates that there must be a certain magnitude in the imbalance of the balance fluxes on either side of the divide before the divide responds to this. It also highlights the importance for fine mesh resolution to resolve these processes. In our simulations, first order convergence between the different mesh resolution (down to ∼300 m) was absent, indicating that

even higher mesh resolutions may be required to achieve this asymptotic behaviour. A flux analysis could not be performed for Söråsen because the selected model domain does not completely cover the ice rise.

### 5.3 Ocean perturbation induced divide migration

Since divide migration amplitude is ≤1 ice thickness away from the initial divide position and the mean migration rates are <0.75 m/yr, we interpret this as shelf thickness perturbations resulting in slow divide migration. Especially in the context of

complete shelf removal being a rather extreme perturbation, the intermediate thinning scenarios might provide a more realistic experiment for the recent past of the Ekström catchment. The simulated small migration amplitudes for the intermediate shelf thinning scenarios (<300 m) indicate that due to the wedged-in geometrical setting of Ekström Ice Shelf, a large portion of the shelf thickness needs to be removed before any flux increase across the grounding line becomes apparent and leads to migration of the divide. This means that shelf thickness perturbations in our experiment would most likely result in a left tilted

Raymond stack, rather than leading to an abandoning of the initial Raymond stack (Figure 2f).

Based on our asymmetric buttressing hypothesis, a simple interpretation of our results would be that Jelbart Ice Shelf for Halvfarryggen and Ekström Ice Shelf for Söråsen provide more buttressing than their respective counterparts in the west, as the divides of both ice rises migrate to the east. However, when using Schoof's flux formula (Schoof, 2007) together with the computed initial fluxes to estimate buttressing ($\Theta$) for Halvfarryggen, the derived values for $\Theta$ are similar for both shelves

(Table 3). Despite the similar stress reduction through thinning or removal of the ice shelf, increase of absolute flux across the grounding line differs. This asymmetry is not induced by asymmetric buttressing, but is caused by the difference in initial flux across the grounding line, which is almost an order of magnitude higher in the east than the counterpart in the west. If now the stress is reduced by the same finite amount, the flux imbalance between east and west will widen (Table 3), resulting in the divide to migrate to the west. We infer from our model simulations that while buttressing induces divide migration, it is

by no means necessary to have asymmetric buttressing for the divide to migrate. The more important determining factor as to how far the divide is going to migrate is the absolute flux imbalance between the two sides of the divide. If we use the flux imbalance from the three different shelf thinning/removal simulations (Runs 8-10, Table 2), the relationship is almost linear between flux imbalance and the resulting divide migration. If this linear fit equation and the modelled flux imbalances are used to predict divide migration for the same three simulations, migration amplitudes of -718 m, -277 m, and -16 m are predicted.

This compares reasonably well with the computed divide migration amplitudes after 100 years of -734 m, -235 m, and -43 m, respectively. This does not mean that this relationship must be linear, but underlines the fact that flux imbalance is much more important than the buttressing provided by the ice shelf for divide migration.





**Table 3.** Flux calculations, derived buttressing factors, and stress reduction calculations for both sides of the divide for Halvfarryggen ice rise from Runs 8-10 (Table 2). GL flux reduction is computed by dividing the $\Delta$GL Flux (column 3) by GL Flux (column 2) and the buttressing factor and stress reduction are calculated from Schoof (2007) (eq. 29).

| Simulation | GL reference flux [$\times 10^8$ m$^3$yr$^{-1}$] | GL flux [$\times 10^8$ m$^3$yr$^{-1}$] | $\Delta$GL flux [$\times 10^8$ m$^3$yr$^{-1}$] | GL flux reduction [-] | Derived buttressing factor $\Theta$ [-] | Stress reduction [%] |
|---|---|---|---|---|---|---|
| No shelf east | 9.553 | 23.65 | 14.01 | 0.592 | 0.21 | 79.0 |
| No shelf west | 1.874 | 5.227 | 3.354 | 0.642 | 0.18 | 81.0 |
| 50% shelf east | 9.553 | 15.21 | 5.657 | 0.372 | 0.36 | 64.0 |
| 50% shelf west | 1.874 | 2.957 | 1.083 | 0.366 | 0.36 | 64.0 |
| 90% shelf east | 9.553 | 10.96 | 1.407 | 0.128 | 0.60 | 40.0 |
| 90% shelf west | 1.874 | 2.315 | 0.441 | 0.191 | 0.52 | 48.0 |

### 5.4 Comparison SMB and ocean induced divide migration

Despite the fact that the SMB perturbation represents the more physically realistic perturbation than the most extreme shelf thickness perturbation (shelf removal), the SMB perturbation results in fast divide migration and the shelf thickness perturbation leads to slow divide migration. This has different consequences for the geometry of the Raymond stack, with fast migration leading likely to a Raymond stack abandoning and slow migration resulting in a left tilted Raymond stack (Figure 2e,f).

The response of the divide position to ocean perturbations is primarily controlled by the subglacial topography with lateral buttressing only being a controlling factor of second order. The modelled short-lived response of the increased grounding-line flux to all ocean perturbations is typical of drainage basins located on prograde sloping bedrock, where the instantaneous removal of all buttressing leads to a sudden, but short-lived response. Similar results have been obtained from modelling of ice-shelf collapse in the Antarctic Peninsula region (Schannwell et al., 2018).

As both ice rises in the model domain, and to the authors' knowledge most other ice rises around Antarctica as well, are located on subglacial topography plateaus, the potential for grounding-line retreat is limited. Because ice flux across the grounding line is primarily a function of ice thickness at the grounding line (Schoof, 2007), the initial retreat of the grounding line on prograde slopes often leads to thinner ice at the grounding line and in turn leads to a reduction of ice flux across the grounding line that can even be lower than before the perturbation. Similarly, the response of the divide position to SMB perturbations also seems to be primarily controlled by the subglacial topography with the magnitude of the flux imbalance between east and west of the divide being a controlling factor of secondary order. Evidence for this is provided by the unadjusted SMB perturbation (Run 5, Table 2) and the SMB grounding-line flux perturbation (Run 6, Table 2) simulations, which both converge to the same new steady-state divide position, even though the forcing is different by a factor two.

As SMB perturbations directly affect surface topography, this type of perturbation leads to quicker response times of ice-divide migration in comparison with shelf thickness perturbation with e-folding times of 95-170 years and 30-50 years for shelf





thickness and SMB simulations, respectively. This means that not only is the magnitude lower, but also the timing of ice-rise divide migration is delayed in the case of shelf thickness perturbations. Moreover, even though the magnitude of the initial perturbation is lower for the SMB simulations, divide migration is larger by a factor ∼3.4 and ∼3.9 for Halvfarryggen and Söråsen, respectively. The divide-migration rate and amplitude for SMB perturbations is most likely heavily dependent on the

spatial pattern of the perturbation, with SMB perturbations near the divide likely leading to faster and larger divide migration than SMB perturbations that have their maximum farther away from the divide (Hindmarsh, 1996).

However, the divide-migration rates computed for Halvfarryggen and Söråsen from shelf thickness perturbations are still important for the analysis of the geometry of Raymond stacks. The magnitudes are of similar amplitude to divide migration rates inferred for Siple Dome (Nereson and Waddington, 2002). Moreover, Ekström Ice Shelf and Jelbart Ice Shelf belong to the

smaller ice shelves around Antarctica, making it likely that ice rises with larger ice flux across the grounding line in combination with larger buttressing provided by the surrounding ice shelves may well cause larger divide migration rates.

The rate of triple junction migration appears to be closely linked to the migration rate of the main divide arm and thus seems to be similarly susceptible to migration than divide ridges. This bears importance for selecting potential ice-core drilling sites on these type of ice domes and a tilted Raymond stack may indicate a displacement of the triple junction as well.

## 15   5.5   Model limitations

By calibrating our ice-sheet model on the Ekström Ice Shelf catchment, we aim to introduce commonly employed initialisation techniques in large-scale ice-sheet modelling to ice-rise modelling. The advantage of the calibration is that buttressing is simulated in a realistic fashion. Without the calibration, large thinning/thickening rates would result in unrealistic model results. However, the calibration matches observed horizontal velocities with modelled horizontal velocities without any constraints on

vertical ice velocities. This leads to the situation that any errors in the horizontal velocities propagate into the vertical velocity through mass conservation. As horizontal velocities in the divide region are close to zero, small errors in horizontal velocities have a large effect on vertical velocities and thus the reconstruction of Raymond stacks. In addition, due to computational constraints, only 10 equally spaced vertical layers could be employed. For an ice thickness of ∼900 m at Halvfarryggen ice rise, this corresponds to a vertical resolution of ∼90 meters. While this vertical resolution is sufficient for our ice-rise divide

migration purposes, a much higher vertical resolution (∼30-40 layers) would be necessary to model Raymond arches at the required detail (Drews et al., 2015). Similarly, despite refining the mesh locally down to 300 m, we do not observe first order convergence in the ice-divide migration amplitude as we refine the mesh. This means that that the difference between migration amplitudes at 1 km and 0.5 km horizontal resolution should be half of the difference between 2 km and 1 km. While this does not affect the results of the paper, it indicates that even higher mesh resolutions ought to be employed and that finer meshes

than presented here (<300 m) may result in larger divide migration amplitudes.

In spite of the advanced ice-sheet model employed, compromises in the complexity of the experimental setup had to be made to make these simulations computationally feasible. These simplifications or approximations were done with the goal of focussing on ice-rise divide migration at the expense of accurately simulating Raymond arch formation. In the following, we will list these simplifications and regard each of them as future avenues to improve on the simulations presented here. As suggested



by many previous ice-rise divide studies, the commonly used exponent of the ice rheology law (n=3) is not able to reproduce the Raymond arch amplitudes from observations, but often a higher exponent (ni≈4.5) is chosen that better matches the arch amplitudes from observations (Martín et al., 2014; Drews et al., 2015; Bons et al., 2018). Moreover, Martín et al. (2009a) showed that the commonly employed approximation of ice being an isotropic material in large-scale ice-sheet models is not

valid at ice divides, where a preferential orientation of the ice crystals leads to enhanced ice deformation. Changes in ice deformation can also be caused by changes in ice temperature, where warmer ice leads to enhanced ice deformation and cold ice reduces ice deformation. While the effect of temperature and enhanced/decreased ice deformation could introduce differences in divide position, to the author's knowledge no one has comprehensively shown this. However, previous studies have found that thermomechanically coupled models exhibit warmer ice at the base under the divide (Martín and Gudmundsson, 2012),

which could potentially indicate that divide migration may occur faster. As our model is not thermomechanically coupled, these effects are ignored in the simulations. Even though ice temperature and anisotropy have been identified as important parameters to be able to reproduce the internal structure of ice divides, it is still uncertain as to how much they affect divide migration.

We performed our simulations with only one type of sliding law (linear Weertmann), without testing alternative implementa-

tions. Even though other modelling studies have shown that this type of sliding law generally results in smaller grounding-line retreat than other sliding laws (e.g. pressure-limited sliding law; (e.g. Schannwell et al., 2018)), it remains difficult to assess the importance of the sliding law for divide migration. On the one hand, reduced basal drag may lead to enhanced grounding-line retreat (Price et al., 2008; Gillet-Chaulet et al., 2016), but on the other hand ice near the divide region might be frozen to the bed and sliding can be neglected in these areas. Therefore, we believe that the choice of the sliding law is likely to have a limited

impact on our results. Given that many previous ice divide modelling studies assume no basal sliding at all (e.g. Martín et al., 2006), the largest impact of this simplification will be on the grounding-line position in the ocean perturbation simulations. But even there, owing to the prograde sloping subglacial topography, the effect of different sliding laws should not be a major concern for the computed divide migration rates.

## 6 Conclusions

We used a calibrated 3D ice-sheet model including grounding-line dynamics and shelf flow for the Ekström catchment to investigate the coupled transient response of ice-rise divides and triple junctions to perturbations in the SMB and ice-shelf thickness. Our perturbation simulations for the Ekström catchment reveal that SMB perturbations result in fast divide migration (up to 3.5 m/yr), while shelf thickness perturbations only trigger slow divide migration (<0.75 m/yr). The amplitude of divide migration is predominately controlled by the subglacial topography and SMB with ice-shelf buttressing being of secondary

importance.

We find in our simulations that asymmetric buttressing is not a required condition for ice-rise divide migration, but rather as to how much the divide will migrate is determined by the flux imbalance between either side of the divide. Both ice rises show a closely coupled response to the perturbations with divide migration being similar in timing and magnitude. Based



on our simulations, the geometry of the Raymond stack could provide clues about the forcing mechanism behind the divide migration, with an abandoned Raymond stack being more likely linked to SMB perturbations. For tilted Raymond stacks, either a smaller SMB perturbation than prescribed here or shelf thickness perturbations are equally likely. This separation of the trigger mechanisms for different types of Raymond Arch geometries is integral to fully unlock the potential of ice rises as

ice-dynamic archives, potential ice-core drilling site, and to better constrain paleo ice-sheet models.

We find that a high mesh resolution (<500 m) is required in the vicinity of the dome to capture ice-rise divide migration at the desired detail, as maximum migration amplitude is <4 km in our perturbation experiments. To avoid unrealistic ice mass loss in transient simulations around the divide region, where longitudinal and bridging stresses are important, the same force balance approximation (e.g. FS for ice divides) should be used in the initialisation and forward simulation of the ice-sheet model.

Finally, migration of the triple junction closely follows the migration pattern of the main ridge, which may proof useful in the future selection of ice-core drilling sites. For example, in the absence of divide migration records, the migration history of the triple junction could be used as proxy to locate the best ice-core drilling site. The model setup is suitable for glacial/interglacial simulations on the catchment scale, providing the next step forward to unravel the ice-dynamic history stored in ice rises all around Antarctica.

*Author contributions.* CS and RD conceived the study with input from OE, TE, and CM. Simulations were run by CS with assistance from FGC. The manuscript was written by CS and RD and all authors contributed to editing and revision.

*Competing interests.* The authors declare that there are no competing interests

*Acknowledgements.* We thank V. Helm for providing us with the TanDEM-X digital elevation models. CS was supported by the Deutsche Forschungsgemeinschaft (DFG) in the framework of the priority programme "Antarctic Research with comparative investigations in Arc-

tic ice areas" by the grant MA 3347/10-1. The authors gratefully acknowledge the compute and data resources provided by the Leibniz Supercomputing Centre (www.lrz.de).



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
