# Peer review of "Kinematic response of ice-rise divides to changes in ocean and atmospheric forcing"

_The Cryosphere, 2019_

## Referee Comment (RC1) · Anonymous Referee #1 · 10 May 2019

This study is an advance in modeling ice-rise evolution and sets a new state-of-the-art from which to understand more. The modeling work is clearly a strong contribution and it is interesting to contrast the response of the ice-rise divide to surface mass balance and to oceanic forcing. My overall reaction is that a stated goal of advancing modeling capabilities was to be able to interpret ice-rise stratigraphy as a function of the history of forcing (based on the abstract and introduction). The modeling shows many cases that indicate how forcing may be imprinted on the ice rise, but doesn't focus directly on the stratigraphy. The discussion mentions general features of the Raymond stack in relation to these calculations but doesn't address how specific histories may be inferred, and even if that goal is now accessible because of this type of advance

in modeling or if data limitations are still significant. In particular, I couldn't connect all the results about divide migration to the overall goal of inferring past forcing and the resulting ice-rise evolution (including divide migration) without relating what is possible and what has been recorded through the stratigraphy. If the main point is how much the divide and triple junction may migrate at all, then the motivation could be reworked to emphasize that the timing and magnitude of migration is something that we need to know, maybe just even for the overall evolution and stability of ice rises less than the imprint on stratigraphy and how that history may be inferred in future work. The conclusions seem to summarize best the main takeaways and it would help if the abstract, introduction, and figures better guide the reader through all of the model results in a more cohesive way to showcase these key points.

Also, if this type of work on triple junctions is completely new then I would highlight that more. I think that it is because 3D models have not been applied like this to ice rises before – so, what do these results mean for 2D interpretations? Are triple junctions commonly observed on ice rises?

I think the conclusions and outcomes of this work would be stronger if the authors can better bridge between what has been done here, and where this can go interpreting ice-rise data and/or understanding ice-rise evolution in general. As a step towards making this point more clear in the text, it may be necessary to rework figures and/or text so that these main messages are clear and that the information shown in the figures is understood to be supporting a particular overall result (or results), as well as displaying specific calculations. As it is now there are a lot of specifics presented and it is hard for the reader to know the best way to use all of the results outside of this work. But, it is clear that the work is a strong contribution and hopefully this feedback can help to strengthen the presentation and therefore the impact of these results within the wider community.

Specific comments:

Pg. 1, Line 4: "other archives are missing" – if space, I'd be more specific about what archives you are referring to that are currently unavailable

Pg. 1, Line 18: Clarify that ice rises are independent of the main ice sheet but what seems important here is that they are isolated from the ice shelf

Pg. 2, Line 18: Suggest rephrasing "It appears higher Glen flow indices than n>3" since that is redundant and not as directly written as could be

Figure 1: Seems like it would be more clear in the figure to use smaller dots to represent ice-rise locations

Figure 2: Type that is in dark-colored brown parts of the figure cannot be seen very well. I printed in black and white and it is unreadable. It is a matter of preference, but I think it helps the reader to have a), b), c) listed before you say what they show, rather than after.

I'd be clear in what you are showing for perturbed cases in Figure 2 that these are isochrones and geometry for new steady state subject to these perturbed conditions

Would be worth more fully referring to Figure 2 in the text, as it isn't completely clear how much of a cartoon this is vs. an illustration of the two cases you'll try

Pg. 4, Line 11: Are these ice shelves larger than typically found or can you qualify "large" here for better context?

Pg. 4, Line 15: in this question does it have to be "or", could it be "and" among all three controls that you mention?

Pg. 4, Line 24: I'm not sure what is meant by "…belong to the larger ice rises in Antarctica…" – but here is where you give context to size in relation to other ice rises, maybe worth mentioning earlier?

Equation 1: Isn't there a minus sign missing?
Pg. 6, Line 7: Is there a physical meaning to the tuning parameters that can be shared simply here without having to go back to Favier et al. (2016)?

Pg. 7, Line 16: Check formatting of ; and )

Pg. 7, Line 23: Need to fix so that subscripts are for both B and C for each parameter

Pg. 7, Line 25: I'm not sure that I'd call L-curve analysis a way to "calibrate" the regularization parameters, really it is a way to pick them following a set of assumptions

Pg. 8, Line 2: What do you mean by "data inconsistencies"?

Also, would be good to clarify that the simulation length you are referring to is the relaxation simulation (10 years), as in the table you show 1000 years

Figure 3: Axes labels and text in figures a) and b) are small and hard to read without zooming in; colors for velocity misfit are hard to map back to the colorbar because the circles are small

Would be worth discussing if these misfits are reasonable and how that is evaluated, not just that the misfit is minimized without overfitting

Pg. 8, Line 10: Are you referring to the magnitude of the SMB? It isn't clear in this sentence

The following point "…we adjust the SMB using the observed model drift following the relaxation simulation" also isn't clear to me what you have done, would be good to elaborate more and to explain better why this is reasonable to do

The following sentences are also not clear to me, especially "we treat the unadjusted SMB as a simulation with a perturbed SMB" – are there some words missing?

Table 2: Is there a misplaced mention to "Run 6", or what does SMB forcing mean here?

Figure 4: In this case the text fonts are so big it is almost distracting to what you are

[Figure]

trying to show (but readable!). Mesh elements should be two words.

Pg. 10, Line 3: I know that it is used, but "inverted basal drag coefficients" sounds funny, so maybe state that these are found by solving an inverse problem

Do these two cases use the same regularization parameters? Is an order of magnitude difference as significant as it sounds?

Pg. 10, Line 8: Would rephrase "in divide proximity" to be "in the proximity of the divide", or even better "near the divide"

Pg. 10, Line 9: What should the reader take away from the statement "Thinning rates were much smaller when using the FS forward model" – what lowering rate was estimated and how did you know that was "good enough"?

Figure 5: Panel b) title should be "FS" and not "NS"

Is there no way to have these on the same color scale? Or, use a different color range as it is just too tempting now to compare them side-by-side

Axes labels are too small to read.

Pg. 11, Line 5: should be "exceeds"

Pg. 11, Line 7: Would be helpful to say the duration of the simulation here (1000 years?)

Pg. 12, Line 9: By "disparity" used here do you just mean "difference"? Is there more to say about why the flux is so different between east and west, other than that was the forcing that was setup?

Figure 6: I spent a long time trying to figure out what is plotted here, so for what that is worth it may be better to plot and/or describe this differently. - the concept of a swath profile wasn't completely clear, or at least would be good to explain more about why the swaths were chosen in these locations - I didn't understand what was meant in the

caption by "backward migration" - Would help to explain more about what the values of balance flux mean in relation to understanding about how different forcing imprints a different history on the ice rise - axes labels are too small

Also, why have these four panels together, would it be better as two figures with two panels each? Or, explain more what we learn by looking at the time series of balance flux as I had a hard time connecting to the point there.

Figure 7 was also hard to take in all the information shown, especially with such a bright color scale I had to zoom more to see the lines and try to relate them back to the different runs and what was shown in the other figures. I appreciate this is hard to plot, but more text around what you are plotting – and why – would help.

Also, how and why were "selected times" chosen?

Figure 10 also took awhile to work through. Some comments: - "total mean divide migration" isn't clear how this was calculated, especially vs. "mean divide migration" - axes labels are too small - the units of the GL flux perturbation aren't intuitive – is the relative difference what is important here?

Pg. 21, Section 5.2: I'm sorry if I lost the point here, but is these ice rises have been stable for 9ka then why investigate divide migration here over 1kyr timescale? It would be helpful to connect what you are constraining about this specific site's history to all the calculations that have been done investigating generalized forcing. I guess that I thought some of these cases may have happened here, but if not that should be really clear (and sorry if I missed it)

Pg. 24, Line 10: Do you mean "cause" larger divide migration rates, or that these configurations could experience larger migration rates?

Pg. 26, Line 10: Should be "prove useful"

As a general question, is this work all about understanding past behavior, or can this understanding of how physical mechanisms drive divide migration inform us about the

sensitivity of ice rises and possibly some ice rises that have configurations that make them more vulnerable to ungrounding. Or, is the divide migration focused on here not significant enough to affect ice-rise stability?

---

## Referee Comment (RC2) · Anonymous Referee #2 · 10 Jun 2019

**1 Overview:**

As the title states, this work attempts to understand the kinematic responses of ice-rise divides to changing oceanic and surface mass balance (SMB) forcing, with an aim to understanding the causes of past ice-rise migrations evident in observations of isochrone patterns in existing ice rises. The authors initialize their model to match present-day conditions for two East-Antarctica ice rises, and then perform a set of numerical experiments to examine the effects of surface-mass-balance (SMB) forcing and ocean forcing (via ice-shelf thinning). the experiments are well-conceived and the results show an interesting differentiation in the rates of response as seen in the icedivide positions. This is a nice piece of work which deserves publication after some issues have been resolved.

My biggest concern is that the simulations are under-resolved. In fairness, the issue is mentioned in the text, but mostly in passing, when in reality, under-resolution has the potential to call all of the results in this work into question. It's clear that the "regular"-mesh runs are under-resolved, given the major differences between the "regular" mesh and the "refined" 500m one. The 350m mesh looks promising, but you need another data point to demonstrate that you're in the convergent regime, since the regular→500m→350m runs don't appear to show any sort of consistent trending behavior (I'm specifically looking at the long-term behavior in figure 6a here – assuming I'm reading the results correctly, the trend from "regular"→500m is to reduce displacement, then the trend from 500m→350m is to increase displacement, so there's not much of a consistent convergence signal). Ideally, you'd run one demonstration run finer than 350 m which would reinforce the trend from 500m→350m and would demonstrate that the 500m mesh is sufficiently resolved to capture the same dynamics as the more-refined solutions. Otherwise, you really don't have a lot of confidence that you're entering the asymptotic regime. I do realize that might be computationally unattainable. A shorter test run may well be sufficient to make this case.

In all honesty, there doesn't seem to be much point in spending as much time and space as you do on the "regular" mesh results, since they are so clearly under-resolved as to be of dubious value.

I'm also concerned about trying to glean so much information from ice divide positions when much of the response is distances which are less than a single mesh cell. Is there perhaps another quantity which might be useful to reinforce your conclusions? In modeling, integrated quantities are often more useful for filtering out any mesh-dependent noise. Perhaps some sort of weighted moments of the ice thickness or patterns of changes in ice thickness would be useful here.
**2 Specific points:**

1. Figure 1: You should note in the caption that the inset figure (b) is rotated with respect to the full-continent figure (a).

2. p2, line 9; "ice-dynamic archive" should be "archives"

3. Figure 2: It would be helpful to point out that subfigure (e) doesn't necessarily correspond to subfigure (b) and subfigure (f) doesn't necessarily correspond to subfigure (c), although the layout encourages that assumption.

4. p3, lines 6-9. It would he helpful if these two sentences were re-ordered to match the ordering in figure 2 (e-f). (fast migration, then slow)

5. p 4, line 2: "relict" -> "relic"

6. p4, line 24: "km2" should be $km^2$ (2 should be an exponent)

7. p4, line 32: "... ice shelves receive..." – should it be "ice shelves provide"?

8. p5, line 5: (possibly too pedantic on my part), I'd suggest "a complete description" instead of "the most complete...". Also, I'd suggest changing "FS flow model" to ' "full-Stokes" (FS) flow model' for accessibility.

9. p7, line 1: I'd suggest changing "sea pressure is prescribed" -> "hydrostatic sea pressure is prescribed"

10. p7, line 21: you have "$J_m$" twice instead of "$J_m$" and "$J_p$". Likewise, line 23 has two $\lambda_C$'s and two $J_C^{reg}$'s

11. p8, "Experimental design" –

    (a) Can the grounding lines move/retreat in your model, or are they held fixed?

(b) How do you handle the 1km or so of remaining ice shelf once you remove the downstream shelf in the shelf removal experiments? For example, is the shelf thickness maintained at the original thickness profile? If it is allowed to change, what constraints on shelf thickness are maintained? (I could, for example, envision a response to downstream shelf removal in which velocities in the shelf remnant (and in the upstream grounded ice) increased, causing an increase in shelf flux, which could lead to thinning and grounding-line retreat.) Also, what forcing (subshelf melt + SMB + calving rules) do they see?

12. Figure 5(b): Caption above should be "FS friction coefficient", not "NS" (unless you're actually solving the Navier-Stokes equations)

13. Somewhere in the problem description, it would be useful to show the bedrock geometry you're using, to make it apparent that the ice rises seem to be on non-retrograde bed slopes, for example.

14. Somewhere in the text, clearly describe how ice divide positions are computed. You refer to the swath, and I think you're computing averages over the swath, but it's not clear how that's done. In particular, I'm more than a bit concerned that the changes in position that you're reporting are less that a single mesh spacing. How long is the swath? Do you see evidence that the ice divide could be rotating relative to the swath-normal direction?

15. Figure 6:

(a) The figures are too small to be legible on a printed page – they're only usable by zooming in on the electronic version. Please make them larger (perhaps a 2x2 layout). thicker lines would help as well. Please ensure that the printed-page version of the figure is usable.

(b) In subfigure b, there is a jump in the displacement followed by a retreat around 350-400 years which occurs at the same *time* in the different-resolution runs (vs. at the same displacement location), which seems to indicate that it's being driven by something in the external flow. Can you comment on that? Do you have any idea what's causing it? It seems unlikely to be simple noise since it shows up in more than one experimental run.

(c) Subfigures c and d demonstrate conclusively that the "regular" mesh is under-resolved. Could you include the results from the 350m run on subfigure (d)? If they're similar enough to the refined 500m results, they would bolster the case that the 500m results are useful.

16. Figure 7: It's not clear how useful showing "regular" mesh results is, since they're demonstrably under-resolved.

17. Figure 8:

(a) As with figure 6, these plots are unreadble on the printed page. A 2x2 layout would probably be more useful here as well.

(b) Could you do a 350m finest-resolution run for the ocean-forcing (shelf removal) case as well?

18. Figure 9: As with Figure 7, it's not clear how useful showing the "regular mesh" results is.

19. Figure 10:

(a) I think you've mislabled the two 90% lines? (the trends would make more sense if the East-90% and West-90% lines were swapped). If that's not the case, then it would be useful to swap them anyway and have line-color denote forcing amount, and solid vs. dashed represent east-west.

(b) Could you increase the vertical size of subfigure c? It's hard to discern what's going on after 5-6 years, particularly whether the lines stay above y=0. Additional stretching of the plot in the y-direction would definitely help here.

20. p19, line 8: "appears more less distinct" – presumably it's either more or less, unless you're aiming for a second career in politics.

21. Figure 12:

   (a) Any idea what's causing the Cartesian-mesh-like artifacts in the $|\nabla(aspect)|$ fields? I find them puzzling since you're using an unstructured mesh and they seem to be definitely some sort of Cartesian grid artifacts.

   (b) The jumps in subfigure f could be numerical noise on the order of the mesh spacing, couldn't they?

22. p 21, section 5.2. You do a very nice job here of discussing the resolution issue. I'd suggest again that there's not much point in discussing specifics of the "regular" mesh results, other than to reinforce that the 2km mesh is under-resolved.

23. p21, line 27: You spend some time here discussing rates of dynamic response. How do you choose the timesteps for the different runs? Is $\Delta t_{regular} > \Delta t_{refined}$? If you're reducing the timestep for the finer-resolution runs (which is reasonable), could faster dynamic response be a product of finer temporal resolution instead of the finer spatial resolution?

24. p22, line 4. I'd suggest replacing "first order convergence between the different mesh resolutions" with "numerical convergence with mesh resolution", because the issue here is a lack of any convergence at all, not just the inability to obtain first-order convergence (you can have (positive) convergence rates less than first order which are still at least converging)

25. p22, line 5: I think you can confidently replace "may be required" with something like "are likely required"

26. p22, line 23: should "same finite amount" be "same fraction" or "same percentage" or something similar? ("amount" implies a fixed value, like 100m)

27. p22, line 24: "the divide to migrate" should probably be "the divide migrating"...

28. Table 3:

    (a) What is the value in the second column? max GL flux? value at a certain time? Integrated flux over time (in which case the units are incorrect)?

    (b) Column 4 is labeled "GL Flux reduction", but all of the flux values in column 2 seem to represent flux *increases*?

    (c) In the first line of the table, when I subtract 23.65-9.553, I don't get 14.01 as in the table (I get 14.097, which would be 14.10). Am I misunderstanding what's being done here or is this typo?

    (d) Shouldn't the 4th column be relative to the reference flux? (change in flux)/(reference flux) instead of (change in flux)/(new flux)

29. p23, line 6. As mentioned before, if you're making the statement that things are controlled by the subglacial topography, you should show the subglacial topography at some point, preferably with a specific example.

30. p23, line 19: "factor two" -> "factor of two"

31. p24, line: 13: "similarly susceptible...than" -> "similarly susceptile... as"?

32. p24, line 26: "first-order convergence" -> "numerical convergence" or "convergence with mesh resolution"

33. p24, line 27: "that that"

34. p24, line 29: "While this does not affect the results of the paper..." – That's too strong of a statement to make without some proof. You could say something like "while we believe that the dynamic results in this work are still valid..."

35. p26, line 10: "proof" -> "prove"

---

## Author Comment (AC1) · 7 Aug 2019

We thank both referees for their thoughtful and thorough reviews of our paper. We appreciate you taking the time to complete these reviews and welcome your helpful comments. We have revised the manuscript to address your review comments (see below). Throughout this response to review document your (referee review) comments are provided in regular, non-italic font text, our response comments are provided in red font (as here).

**Reviewer 1**

This study is an advance in modeling ice-rise evolution and sets a new state-of-the-art from which to understand more. The modeling work is clearly a strong contribution and it is interesting to contrast the response of the ice-rise divide to surface mass balance and to oceanic forcing. My overall reaction is that a stated goal of advancing modeling capabilities was to be able to interpret ice-rise stratigraphy as a function of the history of forcing (based on the abstract and introduction). The modeling shows many cases that indicate how forcing may be imprinted on the ice rise, but doesn't focus directly on the stratigraphy. The discussion mentions general features of the Raymond stack in relation to these calculations but doesn't address how specific histories may be inferred, and even if that goal is now accessible because of this type of advance in modeling or if data limitations are still significant. In particular, I couldn't connect all the results about divide migration to the overall goal of inferring past forcing and the resulting ice-rise evolution (including divide migration) without relating what is possible and what has been recorded through the stratigraphy. If the main point is how much the divide and triple junction may migrate at all, then the motivation could be reworked to emphasize that the timing and magnitude of migration is something that we need to know, maybe just even for the overall evolution and stability of ice rises less than the imprint on stratigraphy and how that history may be inferred in future work.
The conclusions seem to summarize best the main takeaways and it would help if the abstract, introduction, and figures better guide the reader through all of the model results in a more cohesive way to showcase these key points.
We agree that we don't focus on the stratigraphy. The goal of the paper is three-fold. First, do perturbations in SMB and ocean forcing lead to different divide migration rates? Second, if so, can we infer possible unique Raymond Arch geometries that can be interpreted as past changes in SMB or ocean forcing? Third, is the magnitude of divide migration controlled by the magnitude of the perturbation alone or is it controlled by other factors such as the bed topography?
To highlight these points in the paper, we have rewritten the abstract to make clear that we are really investigating changes in ice-rise divide position in response to SMB and shelf thickness perturbations. Moreover, we have removed redundant internal stratigraphy material from the introduction and sharpened the focus towards divide migration. We have updated the discussion section to underline that this work is a first step towards being able to interpret Raymond stacks with regard to their main forcing mechanism. While we are confident that abandoned stacks most likely indicate a SMB signal, the situation is more complicated for tilted stacks where shelf thickness and SMB perturbations are equally likely trigger mechanisms.

Also, if this type of work on triple junctions is completely new then I would highlight that more. I think that it is because 3D models have not been applied like this to ice rises before – so, what do these results mean for 2D interpretations? Are triple junctions commonly observed on ice rises?

Yes, it is new for a real world geometry. The main point which we now mention more clearly in the introduction is that some observed radar features (e.g. relic Raymond stack in the ice rise flank) could not be explained, but have been hypothesized to be linked to a merging/splitting of triple junctions. However, our simulations were not tailored towards investigating this and the applied perturbations are therefore not strong enough to give a definitive answer. Nonetheless, we have expanded and rewritten the introduction, discussion, and conclusion section to underline this particular point of the paper.

I think the conclusions and outcomes of this work would be stronger if the authors can better bridge between what has been done here, and where this can go interpreting ice-rise data and/or understanding ice-rise evolution in general. As a step towards making this point more clear in the text, it may be necessary to rework figures and/or text so that these main messages are clear and that the information shown in the figures is understood to be supporting a particular overall result (or results), as well as displaying specific calculations. As it is now there are a lot of specifics presented and it is hard for the reader to know the best way to use all of the results outside of this work. But, it is clear that the work is a strong contribution and hopefully this feedback can help to strengthen the presentation and therefore the impact of these results within the wider community.

As mentioned above, we have sharpened the abstract, introduction, discussion, and conclusions sections for better guidance of the reader to the main outcomes. We have also changed multiple Figures and deleted redundant material as suggested by reviewer 2.

Specific comments:

Pg. 1, Line 4: "other archives are missing" – if space, I'd be more specific about what archives you are referring to that are currently unavailable

We added "…such as rock outcrops …"

Pg. 1, Line 18: Clarify that ice rises are independent of the main ice sheet but what seems important here is that they are isolated from the ice shelf

We added "…is independent of the main ice sheet and the surrounding ice shelves…"

Pg. 2, Line 18: Suggest rephrasing "It appears higher Glen flow indices than n>3" since that is redundant and not as directly written as could be

This paragraph has been removed as we are not focusing on arch amplitude matching.

Figure 1: Seems like it would be more clear in the figure to use smaller dots to represent ice-rise locations

We have made them the dots a smaller. However, the purpose of this Figure is to show that ice rises are widespread all across the continent.

Figure 2: Type that is in dark-colored brown parts of the figure cannot be seen very well. I printed in black and white and it is unreadable. It is a matter of preference, but I think it helps the reader to have a), b), c) listed before you say what they show, rather than after.

We have made the brown fields lighter. As it is a personal preference, we would like to keep the a), b), c) ordering as is.

I'd be clear in what you are showing for perturbed cases in Figure 2 that these are isochrones and geometry for new steady state subject to these perturbed conditions. Would be worth more fully referring to Figure 2 in the text, as it isn't completely clear how much of a cartoon this is vs. an illustration of the two cases you'll try

We have updated the Figure caption to make this clearer (see Reviewer 2 comment). We now refer in 5 different places in the text to Figure 2.

Pg. 4, Line 11: Are these ice shelves larger than typically found or can you qualify "large" here for better context?

We added here: "…15th and 16th largest in Antarctica (Matsuoka et al., 2015)". Area numbers are listed in the study area section.

Pg. 4, Line 15: in this question does it have to be "or", could it be "and" among all three controls that you mention?

Yes, it could. We changed to and/or

Pg. 4, Line 24: I'm not sure what is meant by "...belong to the larger ice rises in Antarctica. . ." – but here is where you give context to size in relation to other ice rises, maybe worth mentioning earlier?

Done. See comment above.

Equation 1: Isn't there a minus sign missing?

Yes, of course. Thanks for spotting this.

Pg. 6, Line 7: Is there a physical meaning to the tuning parameters that can be shared simply here without having to go back to Favier et al. (2016)?

We added: "…G, and A are tuning parameters to constrain melt rates at the grounding line and away from the grounding line respectively, and α is a local tuning parameter (Table 1)."

Pg. 7, Line 16: Check formatting of ; and )

Done

Pg. 7, Line 23: Need to fix so that subscripts are for both B and C for each parameter

Fixed.

Pg. 7, Line 25: I'm not sure that I'd call L-curve analysis a way to "calibrate" the regularization parameters, really it is a way to pick them following a set of assumptions

Changed.

Pg. 8, Line 2: What do you mean by "data inconsistencies"?

We added: "… such as differences introduced by differing acquisition dates of ice surface elevation and surface velocity."

Also, would be good to clarify that the simulation length you are referring to is the relaxation simulation (10 years), as in the table you show 1000 years
Added.

Figure 3: Axes labels and text in figures a) and b) are small and hard to read without zooming in; colors for velocity misfit are hard to map back to the colorbar because the circles are small. Would be worth discussing if these misfits are reasonable and how that is evaluated, not just that the misfit is minimized without overfitting
We increased label and text size for a), and b) and added "Velocity misfits obtained with these parameters are of similar magnitude to previous studies (e.g. Cornford et al., 2015; Schannwell et al., 2018)." to demonstrate that our misfits are of similar magnitude to other studies.

Pg. 8, Line 10: Are you referring to the magnitude of the SMB? It isn't clear in this sentence
The following point ". . .we adjust the SMB using the observed model drift following the relaxation simulation" also isn't clear to me what you have done, would be good to elaborate more and to explain better why this is reasonable to do. The following sentences are also not clear to me, especially "we treat the unadjusted SMB as a simulation with a perturbed SMB" – are there some words missing?
We reworded this section to make it clearer what is done here. It reads: "This asymmetric SMB pattern is consistent with observations (Drews et al.,2013), but does not capture the correct magnitudes. Therefore, we adjust the SMB using the computed model drift following the relaxation simulation. This means for the reference simulation, the SMB forcing consists of the RACMO2.3 field plus the computed spatial thickening/thinning rate (model drift) at the end of the relaxation simulation. This approach ensures that the model drift is eliminated and the divides stay at their initial position. Since without this model drift correction, there is a change in divide position, we treat the unadjusted SMB as a simulation with a perturbed SMB."

Table 2: Is there a misplaced mention to "Run 6", or what does SMB forcing mean here?
No, it is all correct. As is written in the text: "To permit a more direct comparison between ocean forcing and SMB forcing, an additional SMB perturbation simulation is performed, where the SMB is unadjusted and the initial grounding-line flux perturbation from the shelf removal simulation on either side of Halvfarryggen is added to the SMB term (Table 2, Run 6). This is done such that the spatial pattern of the SMB remains unchanged, but the magnitude is different by about a factor two in comparison to the unadjusted SMB."
To facilitate connecting this to the correct simulation run, we added a reference to the specific run.

Figure 4: In this case the text fonts are so big it is almost distracting to what you are trying to show (but readable!). Mesh elements should be two words.
We have decreased the font size.

Pg. 10, Line 3: I know that it is used, but "inverted basal drag coefficients" sounds funny, so maybe state that these are found by solving an inverse problem
Changed.

Do these two cases use the same regularization parameters?

Yes, as stated in the Model initialisation section: "The Tikhonov parameter from the SSA inversion was used for the FS inversion as well."

Is an order of magnitude difference as significant as it sounds?

Yes, it is. As stated in the text, in the simulation using the SSA basal friction coefficients, there is a thinning rate of 200 m/century and in the simulation using the FS basal friction coefficient, this trend is absent!

Pg. 10, Line 8: Would rephrase "in divide proximity" to be "in the proximity of the divide", or even better "near the divide"

Changed.

Pg. 10, Line 9: What should the reader take away from the statement "Thinning rates were much smaller when using the FS forward model" – what lowering rate was estimated and how did you know that was "good enough"?

We added that they are <50 m/century for the FS inversion fields. For our purpose, it would not have made a big difference as we correct for this model drift, but the wider implications for realistic projections are that the mechanical forward model and the mechanical model for the inversion should be the same.

Figure 5: Panel b) title should be "FS" and not "NS"

Changed

Is there no way to have these on the same color scale? Or, use a different color range as it is just too tempting now to compare them side-by-side
Axes labels are too small to read.

Increased axes labels and plotted both Figures on the same color scale at the cost of losing some detail in each plot.

Pg. 11, Line 5: should be "exceeds"

This refers back to thickening rates, so we think that "exceed" is correct.

Pg. 11, Line 7: Would be helpful to say the duration of the simulation here (1000 years?)

Added.

Pg. 12, Line 9: By "disparity" used here do you just mean "difference"?

Yes, changed accordingly.

Is there more to say about why the flux is so different between east and west, other than that was the forcing that was setup?

No, it is just a result of the geometry of the ice rise.

Figure 6: I spent a long time trying to figure out what is plotted here, so for what that is worth it may be better to plot and/or describe this differently. - the concept of a swath profile wasn't completely clear, or at least would be good to explain more about why the

swaths were chosen in these locations - I didn't understand what was meant in the caption by "backward migration"

We have added a paragraph to the main text which explains how divide positions were computed and highlight that the plotted values are averaged along the swath profile and why the swath profiles were chosen in the way they are. The paragraph reads: "Divide positions are computed at every timestep along two swath profiles (~8 km and ~23 km for Halvfvarryggen and Söråsen, respectively (e.g. Figure 7)). The shorter swath profile for Halvfvarryggen was chosen to permit a simple flux balance analysis. The initial start point of the divide is the location of highest surface elevation. From this point, the divide is tracked along the swath profile by following the minimum direction of the aspect gradient until the end of the swath. Computed mean divide migration amplitudes are then averages along the swath profiles (e.g. Figure 6)."

We also added an arrow pointing to the start of the "backward migration". It is the point at which the divide migration amplitude in the refined simulations starts to decrease.

- Would help to explain more about what the values of balance flux mean in relation to understanding about how different forcing imprints a different history on the ice rise - axes labels are too small

We plot balance fluxes in order to investigate whether we can see a confirmation of the model results that show the regular mesh does not exhibit this backward migration whereas all more refined mesh simulations do. The plotted balance fluxes confirm this as they are almost equal in the regular mesh simulation, and vary greatly for the refined mesh simulations. Axes labels have been made bigger.

Also, why have these four panels together, would it be better as two figures with two panels each? Or, explain more what we learn by looking at the time series of balance flux as I had a hard time connecting to the point there.

We have changed the layout to a 2x2 format to make the Figure more readable.

Figure 7 was also hard to take in all the information shown, especially with such a bright color scale I had to zoom more to see the lines and try to relate them back to the different runs and what was shown in the other figures. I appreciate this is hard to plot, but more text around what you are plotting – and why – would help.

Also, how and why were "selected times" chosen?

As requested by reviewer 2, we have dropped the 2 km simulations from Figures 7 and 9 and have combined these two Figures to one. To better explain what the Figure is showing, we have expanded the Figure caption and now elaborate on what we mean by "selected times".

Figure 10 also took awhile to work through. Some comments: - "total mean divide migration" isn't clear how this was calculated, especially vs. "mean divide migration" - axes labels are too small - the units of the GL flux perturbation aren't intuitive – is the relative difference what is important here?

Apologies. "Total mean divide migration" was a typo. In all cases we mean "mean divide migration". A brief description of how this was calculated has been added to the manuscript. To highlight the importance of the short time period of the perturbation, we added some text to subfigure c. Axes label's fontsize has been increased.

Pg. 21, Section 5.2: I'm sorry if I lost the point here, but is these ice rises have been stable for 9ka then why investigate divide migration here over 1kyr timescale? It would be helpful to connect what you are constraining about this specific site's history to all the calculations that have been done investigating generalized forcing. I guess that I thought some of these cases may have happened here, but if not that should be really clear (and sorry if I missed it)

Ideally we would have liked to investigate longer timescale than 1kyrs, but due to the high computational costs of the FS model, we are restricted to 1kyrs. The forcing is also not tailored to a specific event e.g. transition from LGM to Holocene, but only looks at the what divide migration rates result from realistic perturbations to the SMB and ocean forcing. From our simulations over 1kyrs for the SMB simulations, we draw the conclusion that if a perturbation like this had happened over the last 9000 years, the Raymond stack would still show it today. As this is not the case, we conclude that a perturbation of this magnitude has not happened over the last 9000 years.

Pg. 24, Line 10: Do you mean "cause" larger divide migration rates, or that these configurations could experience larger migration rates?

The latter. Changed accordingly.

Pg. 26, Line 10: Should be "prove useful"

Fixed.

As a general question, is this work all about understanding past behavior, or can this understanding of how physical mechanisms drive divide migration inform us about the sensitivity of ice rises and possibly some ice rises that have configurations that make them more vulnerable to ungrounding. Or, is the divide migration focused on here not significant enough to affect ice-rise stability

We believe that these mechanisms together with the finding that subglacial topography seems to be a first-order control on divide position stability, definitely means that some ice rises are more stable than others. The model setup is easily extended to other ice rises all across Antarctica and the next step will be to test these findings on other ice rises around Antarctica with different bedrock topography settings and different ice shelf settings.

**Reviewer 2**

1 Overview:
As the title states, this work attempts to understand the kinematic responses of ice- rise divides to changing oceanic and surface mass balance (SMB) forcing, with an aim to understanding the causes of past ice-rise migrations evident in observations of isochrone patterns in existing ice rises. The authors initialize their model to match present-day conditions for two East-Antarctica ice rises, and then perform a set of numerical experiments to examine the effects of surface-mass-balance (SMB) forcing and ocean forcing (via ice-shelf thinning). the experiments are well-conceived and the results show an interesting differentiation in the rates of response as seen in the ice-divide positions. This is a nice piece of work which deserves publication after some issues have been resolved.

My biggest concern is that the simulations are under-resolved. In fairness, the issue is mentioned in the text, but mostly in passing, when in reality, under-resolution has the

potential to call all of the results in this work into question. It's clear that the "regular"-mesh runs are under-resolved, given the major differences between the "regular" mesh and the "refined" 500m one. The 350m mesh looks promising, but you need another data point to demonstrate that you're in the convergent regime, since the regular→500m→350m runs don't appear to show any sort of consistent trending behavior (I'm specifically looking at the long-term behavior in figure 6a here – assuming I'm reading the results correctly, the trend from "regular"→500m is to reduce displacement, then the trend from 500m→350m is to increase displacement, so there's not much of a consistent convergence signal). Ideally, you'd run one demonstration run finer than 350 m which would reinforce the trend from 500m→350m and would demonstrate that the 500m mesh is sufficiently resolved to capture the same dynamics as the more-refined solutions. Otherwise, you really don't have a lot of confidence that you're entering the asymptotic regime. I do realize that might be computationally unattainable. A shorter test run may well be sufficient to make this case.

We absolutely agree that mesh resolution is crucial and also agree that the regular mesh is under-resolved. We also agree that yet a finer resolution would be desirable.
To address this, we performed two additional simulations.

- The first simulation performed the no-shelf simulations at 350 m resolution, and shows that convergence is present albeit not at first order. Please note that the simulation is not finished yet (at ~350 years), but Figure 8 will be updated once the simulation has finished.
- The second simulation we performed used a 250 m resolution at the divide and 2 km elsewhere. However, this simulation only confirms that high resolution in both areas is needed.

We tried to run a simulation with 250 m resolution at the divides and the grounding line. This increases the problem size from 1.3 million nodes to ~4 million nodes, and is beyond the capabilities of our current direct solver setup, as we would need >64 GB memory per node. We have identified this problem and are currently working on an iterative solver setup that will permit higher resolution runs in the future. However, at the moment this is beyond the scope of this manuscript.

Please also note that we are reporting mean values along the swath profile which may not be the best quantity to gauge sufficient mesh resolution. To highlight this, we have added local maximum migration amplitudes which are 2.3 km for Halvfarryggen and 1.3 km for Söråsen in the no-shelf simulations. This corresponds to ~4-7 gridcells for Halvfarryggen and ~2-4 gridcelss for Söråsen, depending on the chosen refined mesh resolution. Therefore, while we might be under-resolving in some areas, this is not the case for the length of the swath profile.

In all honesty, there doesn't seem to be much point in spending as much time and space as you do on the "regular" mesh results, since they are so clearly under-resolved as to be of dubious value.

We agree and have shortened the text regarding the regular mesh in the discussion section. We are now also clearly stating that this mesh resolution is insufficient, especially for the ocean perturbation simulations.

I'm also concerned about trying to glean so much information from ice divide positions when much of the response is distances which are less than a single mesh cell. Is there perhaps another quantity which might be useful to reinforce your conclusions?

This is true for the ocean perturbations and yet higher mesh resolutions would be desirable. That is why we chose to perform an additional no-shelf simulation at 350m. However, anything higher is not possible as we are restricted by the availability of our computational power. We would also like to point at that this is a mean value along a 16 km swath. This means that in some areas there is almost no divide migration and in other areas larger values that the mean (see above). We added a sentence that even higher mesh resolutions may improve our results. It reads:" The low migration amplitudes also show that the employed mesh resolution (~500 m) may be insufficient for the intermediate scenarios, but owing to computational restrictions this is the highest resolution possible."

In modeling, integrated quantities are often more useful for filtering out any mesh-dependent noise. Perhaps some sort of weighted moments of the ice thickness or patterns of changes in ice thickness would be useful here.
We still believe that divide migration amplitude is the most intuitive measure and would like to keep this unchanged.

Specific points:
1. Figure 1: You should note in the caption that the inset figure (b) is rotated with respect to the full-continent figure (a).
Done.

2. p2, line 9; "ice-dynamic archive" should be "archives"
Changed.

3. Figure 2: It would be helpful to point out that subfigure (e) doesn't necessarily correspond to subfigure (b) and subfigure (f) doesn't necessarily correspond to subfigure (c), although the layout encourages that assumption.
We added to the caption: "(e) and (f) are not necessarily the result of forcing in (b) and (c), respectively."

4. p3, lines 6-9. It would be helpful if these two sentences were re-ordered to match the ordering in figure 2 (e-f). (fast migration, then slow)
Changed.

5. p 4, line 2: "relict" -> "relic"
Changed.

6. p4, line 24: "km2" should be km2 (2 should be an exponent)
Corrected.

7. p4, line 32: "... ice shelves receive..." – should it be "ice shelves provide"?
Yes. Changed.

8. p5, line 5: (possibly too pedantic on my part), I'd suggest "a complete description" instead of "the most complete...". Also, I'd suggest changing "FS flow model" to ' "full-Stokes" (FS) flow model' for accessibility.
Changed.

9. p7, line 1: I'd suggest changing "sea pressure is prescribed" -> "hydrostatic sea pressure is prescribed"
Changed.

10. p7, line 21: you have "Jm" twice instead of "Jm" and "Jp". Likewise, line 23 has two λC's and two Jreg's
Apologies. This has been corrected.

11. p8, "Experimental design" –
(a) Can the grounding lines move/retreat in your model, or are they held fixed?
Yes, we added: "In all perturbation simulations the grounding line is permitted to freely evolve."

b) How do you handle the 1km or so of remaining ice shelf once you remove the downstream shelf in the shelf removal experiments?
As stated in the Experimental Design section: "This ensures that the same frontal boundary conditions as for the SMB perturbations still apply to this geometry." The applied frontal boundary condition is listed in the Boundary Conditions sections: "For the ice shelf front boundary, the true vertical distribution of the hydrostatic water pressure is applied and the calving front is held fixed throughout the simulations."

For example, is the shelf thickness maintained at the original thickness profile? If it is allowed to change, what constraints on shelf thickness are maintained?
As stated in the Experimental Design section, only for the intermediate scenario is the thickness of the shelf kept constant. So for the extreme shelf removal simulation, there are no constraints for shelf thickness. If the shelf was thinner than 10 m, the model would keep this 10 m of shelf thickness for numerical stability reasons. As this does not happen in our simulations, we do not mention it in the text.

(I could, for example, envision a response to downstream shelf removal in which velocities in the shelf remnant (and in the upstream grounded ice) increased, causing an increase in shelf flux, which could lead to thinning and grounding-line retreat.)
This is exactly what does happen and what we are investigating in our simulations.

Also, what forcing (subshelf melt + SMB + calving rules) do they see?
Ocean and SMB forcings are summarised in Table 2 for each of the runs. Details are provided in the Experimental design section. As for calving rules, it is stated in the Boundary Condition section that we keep the calving front constant, which means no "calving law" is applied.

12. Figure 5(b): Caption above should be "FS friction coefficient", not "NS" (unless you're actually solving the Navier-Stokes equations)
Changed.

13. Somewhere in the problem description, it would be useful to show the bedrock geometry you're using, to make it apparent that the ice rises seem to be on non- retrograde bed slopes, for example.

We have added a Figure to the discussion section, where we show two cross sections of the subglacial topography of Halvfarryggen and Söråsen, underlining the point of prograde sloping bedrock topography for both ice rises.

14. Somewhere in the text, clearly describe how ice divide positions are computed. You refer to the swath, and I think you're computing averages over the swath, but it's not clear how that's done. In particular, I'm more than a bit concerned that the changes in position that you're reporting are less that a single mesh spacing. How long is the swath? Do you see evidence that the ice divide could be rotating relative to the swath-normal direction?
The reviewer is correct that we computed averages along the swath profile (see above). We have added a brief description of how divide positions are computed to section 4.2. It reads:" Divide positions are computed at every timestep along two swath profiles (~8 km and ~23 km for Halvfvarryggen and Söråsen, respectively (e.g. Figure 7)). The initial start point of the divide is the location of highest surface elevation. From this point, the divide is tracked along the swath profile by following the minimum direction of the aspect gradient until the end of the swath. Computed mean divide migration amplitudes are then averages along the swath profiles (e.g. Figure 6)."
There is no indication of divide rotation. However, even if there was, our algorithm would still find the new divide position. Only the distance computation would become more difficult, but that is not the case in our simulations

15. Figure 6:
(a) The figures are too small to be legible on a printed page – they're only usable by zooming in on the electronic version. Please make them larger (perhaps a 2x2 layout). thicker lines would help as well. Please ensure that the printed-page version of the figure is usable.
We changed the Figure format to a 2x2 layout and also increased line thicknesses to improve readability.

b) In subfigure b, there is a jump in the displacement followed by a retreat around 350-400 years which occurs at the same *time* in the different- resolution runs (vs. at the same displacement location), which seems to indicate that it's being driven by something in the external flow. Can you comment on that? Do you have any idea what's causing it? It seems unlikely to be simple noise since it shows up in more than one experimental run.
Are you talking about the refined mesh simulations? If so, we double-checked, there is no external forcing explaining the features, but since the divide is in some areas less well defined than in others that this might be the reason for this jump.

(c) Subfigures c and d demonstrate conclusively that the "regular" mesh is under-resolved. Could you include the results from the 350m run on subfigure (d)? If they're similar enough to the refined 500m results, they would bolster the case that the 500m results are useful.
We have added the 350m simulation and it indeed shows a very similar pattern to the 500m simulation. This has been added to the discussion section.

16. Figure 7: It's not clear how useful showing "regular" mesh results is, since they're demonstrably under-resolved.
We agree and have dropped the 2 km simulations from the Figures 7 and 9 and have combined these two Figures into one.

17. Figure 8:

(a) As with figure 6, these plots are unreadble on the printed page. A 2x2 layout would probably be more useful here as well.

Changed as Figure 6.

(b) Could you do a 350m finest-resolution run for the ocean-forcing (shelf removal) case as well?

Done

18. Figure 9: As with Figure 7, it's not clear how useful showing the "regular mesh" results is.

See above.

19. Figure 10:

(a) I think you've mislabled the two 90% lines? (the trends would make more sense if the East-90% and West-90% lines were swapped). If that's not the case, then it would be useful to swap them anyway and have line-color denote forcing amount, and solid vs. dashed represent east-west.

Yes, indeed it was. Changed accordingly.

b) Could you increase the vertical size of subfigure c? It's hard to discern what's going on after 5-6 years, particularly whether the lines stay above y=0. Additional stretching of the plot in the y-direction would definitely help here.

We have stretched the subplot c to make the plot clearer.

20. p19, line 8: "appears more less distinct" – presumably it's either more or less, unless you're aiming for a second career in politics.

Yes. Corrected to "less distinct"

21. Figure 12:

(a) Any idea what's causing the Cartesian-mesh-like artifacts in the $|\nabla(\text{aspect})|$ fields? I find them puzzling since you're using an unstructured mesh and they seem to be definitely some sort of Cartesian grid artifacts.

The reviewer is correct. For this plot, we regridded the output onto a Cartesian-grid for this plot. We have added this fact to the figure caption.

(b) The jumps in subfigure f could be numerical noise on the order of the mesh spacing, couldn't they?

Yes, they could. As mentioned above for the divide, the triple junction is also not always very clearly defined. So the high-frequency oscillation should not be overinterpreted.

22. p 21, section 5.2. You do a very nice job here of discussing the resolution issue. I'd suggest again that there's not much point in discussing specifics of the "regu- lar" mesh results, other than to reinforce that the 2km mesh is under-resolved.

Agreed and done. See comments above.

23. p21, line 27: You spend some time here discussing rates of dynamic response. How do you choose the timesteps for the different runs? Is Δtregular > Δtrefined? If you're reducing

the timestep for the finer-resolution runs (which is reasonable), could faster dynamic response be a product of finer temporal resolution instead of the finer spatial resolution?

No this is not the case for our simulations as we always use the same timestep of 0.5 yrs as listed in Table 1.

24. p22, line 4. I'd suggest replacing "first order convergence between the different mesh resolutions" with "numerical convergence with mesh resolution", because the issue here is a lack of any convergence at all, not just the inability to obtain first-order convergence (you can have (positive) convergence rates less than first order which are still at least converging)

We have reworded this section, but we do not agree that we have no convergence at all. For the SMB simulations, we do see better than first order convergence as to our definition of convergence. A clear definition of what we define as first-order convergence has been added to section 4.3.1.

25. p22, line 5: I think you can confidently replace "may be required" with something like "are likely required"

Reworded. We now say that we need mesh resolution of ≤500 m.

26. p22, line 23: should "same finite amount" be "same fraction" or "same percentage" or something similar? ("amount" implies a fixed value, like 100m)

Agreed. Changed to "same percentage"

27. p22, line 24: "the divide to migrate" should probably be "the divide migrating"...

Changed.

28. Table 3:
(a) What is the value in the second column? max GL flux? value at a certain time? Integrated flux over time (in which case the units are incorrect)?

Apologies. We agree this was ambiguous. We added "at year 0" to the column headings where appropriate.

(b) Column 4 is labeled "GL Flux reduction", but all of the flux values in column 2 seem to represent flux *increases*?

Again correct. We mean GL flux reduction in relation to the shelf thickness perturbation simulations. We have added this to the table caption.

(c) In the first line of the table, when I subtract 23.65-9.553, I don't get 14.01 as in the table (I get 14.097, which would be 14.10). Am I misunderstanding what's being done here or is this typo?

Apologies. This was a typo. Double checked all other values, too.

(d) Shouldn't the 4th column be relative to the reference flux? (change in flux)/(reference flux) instead of (change in flux)/(new flux)

No, as we are inferring the flux decrease due to the shelf being present, our shelf thickness perturbation flux at year 0 serves as our "reference" flux.

29. p23, line 6. As mentioned before, if you're making the statement that things are controlled by the subglacial topography, you should show the subglacial topography at some point, preferably with a specific example.
See above. Figure has been added.

30. p23, line 19: "factor two" -> "factor of two"
Fixed.

31. p24, line: 13: "similarly susceptible...than" -> "similarly susceptile... as"?
Fixed.

32. p24, line 26: "first-order convergence" -> "numerical convergence" or "convergence with mesh resolution"
Fixed.

33. p24, line 27: "that that"
Fixed.

34. p24, line 29: "While this does not affect the results of the paper..." – That's too strong of a statement to make without some proof. You could say something like "while we believe that the dynamic results in this work are still valid..."
Fixed.

35. p26, line 10: "proof" -> "prove"
Fixed.